# AI-Enabled IoT for Food Computing: Challenges, Opportunities, and Future Directions

**DOI:** 10.3390/s25072147

**Published:** 2025-03-28

**Authors:** Zohra Dakhia, Mariateresa Russo, Massimo Merenda

**Affiliations:** 1Department of Biology, University Federico II of Naples, 80126 Naples, Italy; zohra.dakhia@unirc.it; 2Department of Information Engineering, Infrastructures and Sustainable Energy, University Mediterranea of Reggio Calabria, 89124 Reggio Calabria, Italy; 3Department of Agraria, University Mediterranea of Reggio Calabria, 89124 Reggio Calabria, Italy; mariateresa.russo@unirc.it; 4HWA srl, Spin-Off Mediterranea University of Reggio Calabria, Via R. Campi II tr. 135, 89126 Reggio Calabria, Italy

**Keywords:** food computing, artificial intelligence (AI), internet of things (IoT), supply chain optimization, food traceability

## Abstract

Food computing refers to the integration of digital technologies, such as artificial intelligence (AI), the Internet of Things (IoT), and data-driven approaches, to address various challenges in the food sector. It encompasses a wide range of technologies that improve the efficiency, safety, and sustainability of food systems, from production to consumption. It represents a transformative approach to addressing challenges in the food sector by integrating AI, the IoT, and data-driven methodologies. Unlike traditional food systems, which primarily focus on production and safety, food computing leverages AI for intelligent decision making and the IoT for real-time monitoring, enabling significant advancements in areas such as supply chain optimization, food safety, and personalized nutrition. This review highlights AI applications, including computer vision for food recognition and quality assessment, Natural Language Processing for recipe analysis, and predictive modeling for dietary recommendations. Simultaneously, the IoT enhances transparency and efficiency through real-time monitoring, data collection, and device connectivity. The convergence of these technologies relies on diverse data sources, such as images, nutritional databases, and user-generated logs, which are critical to enabling traceability and tailored solutions. Despite its potential, food computing faces challenges, including data heterogeneity, privacy concerns, scalability issues, and regulatory constraints. To address these, this paper explores solutions like federated learning for secure on-device data processing and blockchain for transparent traceability. Emerging trends, such as edge AI for real-time analytics and sustainable practices powered by AI–IoT integration, are also discussed. This review offers actionable insights to advance the food sector through innovative and ethical technological frameworks.

## 1. Introduction

The global food sector faces significant challenges as the demand for food increases and the need for sustainability, food safety, and efficiency becomes more pressing. Traditional methods of food production and distribution are increasingly inadequate to meet these growing demands. To tackle these challenges, the integration of modern technologies such as artificial intelligence (AI) and the Internet of Things (IoT) has emerged as a promising solution [1,2]. Food computing, a rapidly developing field, leverages the power of AI, the IoT, and data analytics to enhance food systems across various stages, from production and processing to distribution and consumption. By harnessing AI’s ability to analyze large datasets and the IoT’s real-time monitoring capabilities, food systems can be made more efficient, sustainable, and responsive to dynamic conditions. Together, these technologies have the potential to address pressing global issues like food insecurity, waste reduction, and personalized nutrition [3]. This review paper explores how the convergence of AI and the IoT is reshaping the food sector. We examine their applications in food quality monitoring, supply chain optimization, and sustainable practices. The review also highlights the challenges of integrating these technologies into the food sector, such as data heterogeneity, privacy concerns, and scalability issues, and provides insights into how these barriers can be overcome. Moreover, we critically evaluate the existing challenges in food computing, aiming to provide a comprehensive analysis of the methodologies and strategies used in current research. By focusing on these challenges, this paper intends to guide future researchers and practitioners, helping them focus on the areas that need further exploration and development. This critical review is intended to foster deeper understanding and inspire solutions that drive the advancement of food computing technologies. Thorough literature searches were carried out on academic databases, such as Scopus, Google Scholar, and Web of Science, in order to carefully gather the required information to cover the State of the Art (SotA) of the last four years on this topic. The analysis focused on key terms evaluated between 2015 and 2024, including “food AND computing”, “food AND computing AND AI”, “food AND computing AND AI AND IoT”, and “food AND computing AND AI AND IoT AND device”. After merging the research results from these queries, the analysis concentrated on 160 papers most relevant to the topic. The research trend for food computing papers indexed in Web of Science (WOS) from 2015 to 2024 is depicted in Figure 1, showing a total of 34,653 document findings. The number of publications varied by year, with 1563 papers published in 2015, and a steady increase observed over the following years. In 2016, there were 1720 papers, followed by 2109 in 2017. The number of publications continued to grow, reaching 2335 in 2018. The field saw 2769 papers in 2019, and the number of publications surged to 3286 in 2020. In 2021, there was a significant rise to 4103 papers, followed by 4366 publications in 2022. The trend continued upward in 2023, with 4277 papers, and reached 4814 in 2024. This upward trajectory indicates the growing focus and investment in the intersection of food science and computational methods.

This methodology ensured a thorough examination of the relevant academic literature and provided a strong foundation for integrating insights into the dynamic relationship between AI and the IoT within the context of food computing. The remainder of this paper is organized as follows: Section 2 provides an overview of the background theory essential to understanding food computing; Section 3 presents an in-depth exploration of the State of the Art in food computing, highlighting recent advancements and trends; Section 5 examines the convergence of AI and the IoT in the food sector, discussing their synergistic effects; and Section 7 offers a thorough discussion of the findings related to food computing. Finally, a comprehensive conclusion summarizing the key insights and implications is provided in Section 8.

## 2. Background Theory

The field of food science has long been concerned with the study and management of the substances that provide essential nutrients and energy for the growth, maintenance, and overall health of living organisms. As human societies evolved, so did the understanding and processes surrounding food production, distribution, and consumption. Today, the convergence of technology with traditional food has led to the emergence of food computing an interdisciplinary domain that applies computational tools, AI, and IoT technologies to optimize food-related systems. This approach seeks to address modern challenges in food production, safety, and sustainability, paving the way for innovations in agricultural practices, personalized nutrition, and food supply chain management. The integration of these advanced technologies is reshaping how we perceive food systems, offering new ways to enhance food quality, traceability, and efficiency. In this context, we will begin by presenting the differences between traditional food and food computing, followed by an exploration of the role of AI in transforming food systems. We will then discuss the role of IoT technologies in food-related processes and touch on other key aspects of food computing that are driving innovation and change in the field.

### 2.1. Food and Food Computing

Any material that provides vital nutrients and energy for growth, maintenance, and health that is ingested by living things is referred to as food. This includes both plant-based foods like fruits and vegetables and animal-based foods like meat and dairy [4,5]. In contrast, food computing is an interdisciplinary field that optimizes food-related systems by fusing data science, AI, and food science. The term refers to the application of computational tools to problems in food production, distribution, nutrition, and safety with the goal of improving agricultural practices and enhancing individualized nutrition through data-driven insights [6]. The food value chain spans various stages, from seed planting to consumption, including agricultural production, harvesting, storage, industrial food preparation, processing, distribution, and retail. A diagram of the food supply chain, shown below in Figure 2, illustrates these stages and demonstrates how food computing can be applied across them to optimize processes and improve food safety and sustainability.

### 2.2. Technological Integration in Food Sector

The integration of technologies like the IoT, AI, and big data enhances efficiency, traceability, and food safety throughout the food supply chain. Real-time data from sensors help in quality control and reduce waste, ensuring better management of resources and improved consumer satisfaction.

#### 2.2.1. AI in Food Sector

Food analysis is changing as a result of enhanced data processing made possible by AI and machine learning (ML) algorithms. These systems analyze spectral data for food quality assessment, providing accurate assessments of contamination, freshness, and composition [7]. In recent years, significant progress has been made in using various ML methods for analyzing food quality, such as convolutional neural networks (CNNs) for image recognition and support vector machines (SVMs) for classification tasks. These methods have shown promise in achieving higher accuracy in detecting food quality issues. However, some challenges remain, such as the need for large labeled datasets to train these models, which limits their application in real-world scenarios. When sensor fusion and AI are combined, the accuracy of identifying minute differences is improved [8,9]. Recent studies on sensor fusion have highlighted the importance of combining data from multiple sensors, such as temperature, humidity, and chemical sensors, to improve food safety and quality monitoring. However, one of the key challenges with sensor fusion is dealing with noisy data and aligning data from different sensor modalities. AI streamlines inventory control and delivery in supply chain management, reducing waste. By seeing patterns and abnormalities in enormous datasets, ML is essential to anticipating and averting foodborne illness cases [10]. A variety of ML models, including decision trees, random forests, and neural networks, have been employed to predict and prevent foodborne illness outbreaks. However, one limitation of these models is their dependency on high-quality data from food safety inspections and sensor readings, which are not always available. Future research could focus on improving these models by integrating unstructured data sources, such as social media or consumer complaints, to enhance prediction accuracy. The food sector is reshaped by this synergy, which guarantees efficiency, quality, and safety throughout the production and distribution process [11]. Emerging ML techniques, such as deep reinforcement learning (DRL), could further optimize supply chain operations by improving real-time decision making and resource allocation. However, the application of DRL in the food sector is still at an early stage and requires more exploration. Figure 3 shows how AI is used in the food sector, focusing on real-world examples and important technologies. It highlights applications like smart packaging, food fraud detection, and supply chain optimization. The figure also presents key AI techniques, such as machine learning, predictive analysis, personalized nutrition, smart packaging, and computer vision, showing their role in improving the food sector. Studies in the food sector will focus on integrating novel data sources, such as unstructured data from social media or consumer feedback, to enhance prediction models. Efforts will also be directed towards advancing sensor fusion technologies to address data noise and improve accuracy in food safety monitoring. Additionally, the application of emerging ML models, such as DRL, will be explored to optimize real-time decision making in food supply chains. These advancements are expected to overcome current limitations and drive further developments in the food industry.

#### 2.2.2. IoT Integration in Food Sector

The integration of the IoT in the food sector is the process of implementing networked sensors and devices for automated and real-time monitoring in food systems [12], ensuring the entire supply chain is optimized. The acquisition of data from these devices guarantees quality control and equipment efficiency in food manufacturing [13]. In terms of logistics, it improves transparency and traceability while reducing waste and raising overall supply chain efficiency for food [14]. Figure 4 illustrates the role of the IoT in the food sector, showcasing real-world applications such as crop monitoring and disease detection, precision farming, smart irrigation systems, food traceability, and automated harvesting. It also highlights key technologies enabling these applications, including AI, cloud computing, networking, big data, blockchain, sensors, and the IoT, demonstrating their importance in advancing agriculture and food systems.

### 2.3. Advanced Technologies for Food Processing

AI, ML, and automation are transforming food processing by increasing production efficiency and maintaining consistent product quality. These technologies help to optimize processes, detect defects, and minimize waste, contributing to a more sustainable and efficient food production system.

#### 2.3.1. AI and IoT in Food Sector

The convergence of AI and the IoT represents a significant shift in the food sector [15]. The IoT enhances AI’s capabilities by providing a large number of real-time streaming data, enabling more complex analyses and decision making [16,17]. This synergy is evident in various applications, such as monitoring, where IoT sensors supply environmental data, and intelligent machines that optimize practices by using AI algorithms [18]. Algorithm optimization methods, such as model pruning, quantization, and hyperparameter tuning, can be employed to improve the efficiency and accuracy of AI models. Data augmentation techniques, including image transformations and synthetic data generation, are used to expand the dataset and enhance the generalization ability of AI models in food quality inspection tasks. Transfer learning also plays a key role in improving model accuracy by leveraging pre-trained models on related tasks. These techniques enable more accurate predictions with fewer data, improving model robustness in food-related applications. IoT technologies play a crucial role in the food sector by using a wide range of sensors, including temperature, humidity, gas, and optical sensors, to capture critical environmental parameters necessary for food quality assessment. These sensors enable real-time data collection, which is essential to monitoring food safety, freshness, and storage conditions throughout the supply chain. In food traceability, RFID tags and QR codes enhance transparency and authentication, ensuring the integrity of food products from farm to table. Efficient communication between devices and cloud platforms is achieved through protocols like MQTT, CoAP, and LoRaWAN, supporting real-time monitoring and control [19]. Additionally, edge computing processes data locally on IoT devices, reducing latency and minimizing bandwidth usage, thus enabling faster decision making and operational efficiency in food processing and distribution systems. AI technologies complement the IoT in food computing by providing advanced data processing and analytical capabilities. ML models, including supervised, unsupervised, and reinforcement learning, are used to identify patterns and make predictions related to food safety, quality, and shelf life. DL architectures, such as CNNs and Recurrent Neural Networks (RNNs), are particularly effective in handling complex data types. For example, CNNs are extensively used for image-based food classification, detecting defects, and assessing ripeness levels, while RNNs are effective in time-series analysis in monitoring environmental conditions. Natural Language Processing (NLP) techniques analyze consumer feedback and reviews, enabling personalized nutritional recommendations. Additionally, computer vision systems automate quality inspection processes, significantly improving accuracy and consistency. By integrating these AI techniques, decision-making processes become more data-driven and predictive, enhancing food safety, supply chain efficiency, and personalized consumer experiences in food computing [20]. Global companies have applied these technologies in real-world scenarios. For example, precision irrigation in smart agriculture uses IoT sensors for soil moisture and weather data, while AI algorithms optimize water usage to maximize crop yield. In food processing, AI-driven quality control systems use IoT sensors to detect defects or contaminants in products on production lines, ensuring higher standards of safety and quality. Real-time monitoring in food logistics ensures that transportation conditions meet optimal parameters for perishable goods, minimizing spoilage and waste. By leveraging AI and the IoT together, we can improve supply chain efficiency through real-time tracking in the food sector. Additionally, AI facilitates the analysis of data to gain valuable insights into food quality. When combined, AI and the IoT revolutionize food quality assessment by enabling complex analyses [21]. Figure 5 illustrates the integration of two distinct technologies: the AI-based apple ripeness detection system and the IoT-driven environmental monitoring application. By combining these techniques, the system enables the real-time assessment of both the ripeness of apples and the surrounding environmental conditions. This approach ensures that apples are harvested at the optimal time, considering both their readiness for harvest and the ideal environmental factors, ultimately improving the quality and efficiency of the harvesting process.

#### 2.3.2. Image Recognition and Computer Vision for Food Sector

Systems that are able to recognize and categorize food products from photos are made possible by computer vision techniques, especially image recognition [22]. These systems can compute nutritional values, estimate portion sizes, and automatically detect and label food items by using CNNs. Smart kitchen appliances, food waste reduction programs, and diet-tracking smartphone applications all make extensive use of this technology [23]. Food-related tasks like calorie counting and ingredient identification can be performed more accurately and efficiently due to advanced image recognition models that can differentiate between similar-looking food products and adapt to various lighting conditions [24].

#### 2.3.3. Natural Language Processing for Recipe Analysis and Recommendation Systems

In particular, recommendation systems and recipe analysis depend heavily on Natural Language Processing (NLP) for the processing and analysis of textual food data. NLP techniques extract information about ingredients, cooking times, and nutritional value from recipes to aid in their parsing and understanding [25]. Recommendation systems employ semantic analysis to propose meals according to user choices, dietary constraints, and available ingredients. Additionally, NLP models are employed to create new recipes, offer cooking guidelines, and make tailored meal recommendations. These features improve the user experience by simplifying the process of finding, preparing, and enjoying meals that are customized to meet specific needs [26].

#### 2.3.4. Food Recognition Through IoT and Mobile Applications

Food tracking and recognition are becoming easier with the use of smartphone apps and the IoT. IoT devices use sensors and image recognition to track expiration dates, monitor food inventories, and recommend recipes based on items that are available [27]. Examples of these devices are smart refrigerators and kitchen appliances. Mobile applications frequently incorporate food recognition and computer vision features, enabling users to snap photos of their meals and record nutritional data. These technologies leverage real-time data from linked equipment and user interactions with mobile apps to streamline meal tracking, improve kitchen management, and provide individualized nutrition recommendations [28]. Figure 6 showcases a fruit recognition system that accurately identifies a variety of fruits, including kiwi, pineapple, apple, and strawberries. The system uses advanced image processing and machine learning techniques to classify and distinguish among different fruit types, enabling applications in agriculture, food sorting, and quality control.

### 2.4. Data Security and Privacy in Food Computing

Data security and privacy are critical in the field of food computing because of the large volumes of sensitive data produced by IoT and AI technologies [29]. It is imperative to guarantee the security and integrity of data, especially in applications where real-time monitoring is critical, such as food supply chain management [30]. Strict authentication procedures, safe data storage, and strong encryption techniques are essential to preventing unwanted access [31]. Furthermore, privacy issues are raised during the gathering and processing of private food and health data, making a cautious compromise between using these data for study and preserving individuals’ privacy necessary [32]. As food computing develops, resolving these privacy and security issues is crucial to building industry trust and maintaining moral principles [33]. Privacy-preserving techniques such as homomorphic encryption and differential privacy are gaining traction as methods to secure sensitive food-related data while enabling analytics. Incorporating these technologies into food computing systems could bridge the gap between innovation and compliance with data protection regulations like GDPR [34,35].

### 2.5. Industry Adoption and Challenges

The industry is undergoing significant changes due to the adoption of food computing technologies, where the integration of AI and the IoT optimizes sustainability, efficiency, and quality throughout the supply chain [36]. With the rise of IoT devices, precision farming is becoming increasingly common in agriculture, which leads to improved resource efficiency. Moreover, AI-driven automation is enhancing processes in food processing and manufacturing [37]. However, challenges persist. Widespread adoption is hindered by integration costs, interoperability issues, and concerns related to data security [38]. Additionally, ethical considerations and regulatory compliance regarding data usage pose further obstacles. To fully realize the potential of food computing, the sector must overcome these challenges [39]. Successfully adopting these technologies promises to increase productivity, drive innovation, and pave the way for a more resilient, responsive, and technologically advanced future in many areas of the food sector [40].

Below, Table 1 provides a comparison between traditional food concepts and the emerging field of food computing. It evaluates the involvement of physical properties, computational methods, and advanced technologies like AI and the IoT in each area. This comparison highlights the distinct roles that food and food computing play, as well as their technological applications.

## 3. State of the Art of AI in Food Computing

The integration of AI has become a groundbreaking force in the agri-food sector, driving significant advancements across the entire food supply chain. From optimizing agricultural practices to enhancing food safety and traceability, AI’s potential to revolutionize food computing is becoming increasingly evident. This section delves into the current landscape of AI applications within the food industry, focusing on key innovations that are not only streamlining operations but also enabling smarter, more sustainable solutions. By examining the latest developments in AI, we can better understand how these technologies are reshaping food production, distribution, and consumption in profound ways.

### 3.1. The Current Use of AI in the Food Sector

AI is increasingly being used in the food sector to enhance yield, quality, safety, and traceability. A review article [7] points out that AI can significantly improve food delivery and logistics while reducing waste and streamlining the supply chain. Bannor et al., in [41], offer a bibliometric analysis of AI in food safety, emphasizing that these technologies are powerful tools for enhancing both safety and traceability. Interestingly, Zhang et al., in [42], suggest that by 2023, 77% of consumers would be open to trying products developed by AI, indicating a growing acceptance of AI-generated recipes and food concepts. The role of AI in food biotechnology is further underscored by Barthwal et al. in [43], who discuss innovations like precision fermentation and the development of new process aid enzymes. Companies such as Shiru, Inc., and Vectron Biosolutions [42] are already leveraging AI as a significant force in their product development efforts. Furthermore, Nestle SA has announced plans to invest in AI and ML to enhance its manufacturing processes. Despite the immense potential that AI holds for the food sector, there are still challenges to overcome before it can be widely implemented across various sectors [43]. Figure 7 illustrates an intelligent restaurant system where robotics play a central role in food preparation and delivery. The process begins with an automated food preparation system, where robots handle cooking with precision, ensuring consistent quality and efficiency. After the food is prepared, a robot delivery system takes over, delivering the meal seamlessly from the kitchen to the customer. This innovative system combines AI-driven ordering with robotic logistics to create a fully automated dining experience, improving efficiency, consistency, and customer satisfaction. The entire operation is integrated to offer a smooth, intelligent flow from preparation to delivery, enhancing the dining experience with cutting-edge technology.

### 3.2. Challenges in Food Sector Using AI

In the context of the food industry, the integration of AI and IoT technologies offers significant opportunities but also presents several challenges. These challenges primarily arise from the complexity of managing a large number of data, ensuring data privacy and security, and addressing scalability issues in large-scale operations. To fully leverage the benefits of these technologies, it is crucial to understand and overcome these obstacles through innovative AI-driven solutions and advanced technological platforms. AI-based advances in food computing offer a wide range of benefits, including enhanced efficiency, improved food safety, personalized consumer experiences, and optimized production processes. Technologies such as big data, AI, the IoT, blockchain, and robotics are revolutionizing the food industry. AI and big data collaborate to analyze large datasets for informed decision making, while IoT devices ensure the real-time monitoring of food quality, promoting traceability and compliance with food safety standards. Additionally, consumer feedback is integrated to improve services and products continuously. Digital twins and Augmented Reality enhance the experience by simulating food production environments and engaging consumers interactively. Edge computing enables quick processing and decision making, while blockchain guarantees the transparency and security of food data. Finally, robotics automates food preparation and handling, streamlining operations. Figure 8 below illustrates how these technologies are integrated within the food industry, emphasizing their interconnected roles in improving production, safety, and consumer experiences.

AI-based advances in food computing present a multitude of benefits, but they also present a number of obstacles, such as the following.

#### 3.2.1. Data Quality and Availability

Data availability and quality present significant challenges for AI applications in the food sector. Ensuring that the data used in these applications are accurate and reliable is crucial to making sound judgments and predictions [44]. Figure 9 provides a comprehensive overview of the steps involved in collecting, sharing, and using data. Ontology management plays a crucial role in organizing and structuring food-related data, ensuring semantic interoperability and consistency across various datasets. Ontology-based models provide a standardized vocabulary and well-defined relationships among entities, enabling the seamless integration of heterogeneous data sources, which is essential to AI applications in food computing. These ontologies allow for the classification and categorization of data types, such as nutritional content, food ingredients, agricultural data, and environmental conditions, into a unified framework. Specific examples of ontology management frameworks include Food Ontology (FO) and Agriculture Ontology (AgriOnt), both of which provide structured frameworks for integrating data related to food and agriculture, respectively. The use of ontology management facilitates better data annotation, retrieval, and analysis by linking different datasets through consistent taxonomies. Moreover, ontology-driven systems enable automated reasoning, helping AI models identify patterns and infer new knowledge based on the relationships within the data. These frameworks also aid in addressing challenges related to food traceability, quality control, and sustainability by ensuring accurate and reliable decision-making processes. A thorough evaluation and classification of existing datasets for food computing emphasizes the importance of high-quality data for ML technologies [45]. Furthermore, another perspective highlights the growing importance of data quality in the food sector, suggesting the application of data science methods to enhance the quality, accessibility, and usability of food and nutrient composition data [46]. Additionally, research discusses the critical role of data availability in addressing global food insecurity, particularly emphasizing the use of ML to drive food security initiatives in the absence of primary data [47]. Collectively, these findings underscore the vital importance of data availability and quality in the food sector, food computing, and food security issues that have become even more pressing in the wake of the COVID-19 pandemic, which has heightened the demand for timely and reliable data in this field [48].

#### 3.2.2. Complexity of Food Attributes

AI in the food sector faces challenges due to the intricate nature of food characteristics. Advanced AI algorithms and technologies are essential to analyzing and understanding various attributes of food products, including flavor, texture, and nutritional content [37]. Recent research has explored the relationship between consumer preferences and food complexity, revealing that while highly complex foods can be overwhelming and difficult to grasp, consumers often prefer foods with a moderate level of complexity [49]. Additionally, findings on dietary complexity and satiety indicate that consuming more complex foods can enhance feelings of satisfaction and fullness [50]. These insights suggest that the complexity of dietary features can significantly influence consumer behavior and health outcomes. However, further research is necessary to fully understand the connection between food complexity and various aspects of food intake and health.

#### 3.2.3. Personalization and Cultural Differences

AI in the food sector faces challenges related to personalization and cultural variations. Customizing food products and services to align with individual preferences and diverse cultural traditions requires advanced AI systems capable of understanding and adapting to shifting customer needs [51]. Recent research has provided valuable insights into cross-cultural food preferences and the impact of food culture on multiple facets. Studies have explored the relationships among corporate sustainability, food culture, and cultural inclusion, emphasizing the distinctive regional, social, and cultural attributes of food and its role in reinforcing local identity [52]. A key challenge for AI in food personalization is the complexity of accounting for cultural diversity and dietary variations. AI models must consider a wide array of dietary restrictions, regional food preferences, and cultural practices when making personalized recommendations. These challenges are particularly pronounced in the food sector, where consumer choices are deeply influenced by social norms, historical practices, and individual beliefs. The AI system must not only process personal taste preferences but also navigate this complex cultural landscape, requiring sophisticated algorithms capable of handling these dynamic and often contradictory influences. Furthermore, achieving algorithmic fairness while accounting for such diversity remains a critical obstacle to creating truly inclusive and personalized food solutions. Additionally, ongoing research highlights how food issues and materialities shape food cultures and influence corporate sustainability, bringing attention to the nuanced effects of food culture in various contexts [50]. These findings contribute important perspectives on the role of cross-cultural food recommender systems, as well as the broader implications of food culture in personalization and food computing.

#### 3.2.4. Ethical and Privacy Concerns

AI poses significant challenges in the food sector, particularly around privacy and ethics. Ensuring responsible AI deployment requires a commitment to ethical practices, safeguarding customer privacy, and addressing biases within AI systems [51,53]. Addressing the ethical and privacy implications of food computing is critical. Research highlights the need for stringent protections around personal data and calls for more regulated monitoring technologies to protect civil liberties [54]. Other studies emphasize the importance of developing a shared framework across the food supply chain to address privacy, agency, and accountability [55]. As AI progresses, new legislation is expected to enhance privacy protections, establish risk frameworks, and set standards for auditing data bias and privacy [56]. Data security and corporate monitoring also remain top priorities, as underscored in recent analyses of U.S. cybersecurity and data privacy trends [57]. Overall, addressing ethical and privacy concerns in food computing is essential to ensuring that these technologies are applied responsibly for societal benefit.

#### 3.2.5. Environmental Impact

Addressing the environmental impact of AI in the food sector is essential. Ensuring that the benefits of AI applications in food computing align with goals for environmental sustainability, energy efficiency, and waste reduction is key for long-term viability [58]. The environmental implications of food computing are a growing area of research. Studies show that the food sector is responsible for a significant share of greenhouse gas emissions, with food production alone accounting for 26% of global emissions [59]. However, implementing AI in the food sector to reduce environmental impact comes with certain challenges. One key challenge is the need for accurate and comprehensive data collection, as environmental data from farming, production, and distribution processes can be highly variable. AI models must be trained on large datasets to make reliable predictions, which often requires the extensive monitoring of environmental conditions across various stages of food production. Moreover, the integration of AI with traditional systems in the food sector can be complicated by legacy infrastructure, leading to slow adoption and increased operational costs. Despite these challenges, the potential benefits of AI for reducing food sector emissions and waste are undeniable. Edible packaging offers a promising solution, using fewer resources in production and transport while also helping reduce food waste [60]. Increasingly, food companies are adopting environmental impact reporting, with carbon footprint analysis becoming a central focus. Recognizing both the benefits and challenges of carbon tracking is crucial, as it shapes sustainability practices [61]. The European Food Safety Authority has also initiated efforts to assess environmental impact through a comprehensive Environmental Footprint of Food database [62]. Altogether, these insights highlight the pressing need for the food sector to reassess its environmental footprint and actively work towards reducing emissions and waste.

#### 3.2.6. Regulatory Compliance

Challenges to AI adoption in the food sector often stem from the need to ensure regulatory compliance. For AI technology to align with legal standards, industry protocols, food safety regulations, and data protection laws, careful consideration and adherence to these requirements are essential [63,64]. Regulatory compliance within food computing is complex and requires meticulous attention to established guidelines, including documentation and record-keeping practices for materials that come into contact with food [65]. A critical issue in achieving regulatory compliance is the varying nature of food safety standards across different regions. In some countries, there are discrepancies in the regulatory frameworks that govern food production, data privacy, and AI usage. These differences create obstacles for companies seeking to implement AI technologies across multiple markets, as they must navigate diverse compliance requirements. Additionally, these regulatory inconsistencies can slow the integration of AI technologies, making it more difficult for companies to deploy solutions that can meet the required standards globally. Further insights into the relationship between public enforcement agencies and private parties highlight the intricate dynamics of compliance enforcement in the food sector [66]. To navigate the constantly evolving regulatory landscape, food sectors are increasingly relying on advanced technologies, which enable them to adapt compliance efforts to the latest standards and requirements [67]. These sources collectively underscore the multi-layered nature of regulatory compliance in food computing and the range of approaches being used to ensure adherence.

#### 3.2.7. Integration with Existing Systems

It is challenging to integrate AI technologies with current food production and supply chain management systems [68]. For AI to be successfully adopted in the food sector, it must guarantee smooth integration, interoperability, and no interruption to current procedures [69]. The integration of current systems in food computing has drawn a lot of interest lately, with an emphasis on improving the food sector’s sustainability, efficiency, and safety. To maintain high standards of food safety, for example, the European Food Safety Authority (EFSA) highlights the necessity of collaboration and partnership among all bodies [70]. The journal *Technological Forecasting and Social Change* delves into the most recent research on the digital transformation of food supply chains. It puts forth a framework for implementation that incorporates essential technologies like the IoT to accomplish this goal [71]. Furthermore, these fields of study show the increasing interest in combining current food computer systems to solve the issues of sustainability, innovation, and food safety. The food sector may continue to develop and adapt to the shifting needs of consumers and the environment by encouraging cooperation and the creation of new technology [72].

#### 3.2.8. Real-Time Processing Challenges

In the food sector, real-time processing is becoming more and more important, and integrating IoT technology has a lot of potential, especially when considering business 4.0. The application of the IoT may provide a number of benefits, such as remote monitoring, simulation, and real-time data analysis, which will improve and enhance the food processing industry [73]. Furthermore, the implementation of automated systems and big data in the food supply chain can help with food-related data management, analysis, and reaction, which will ultimately allow for real-time prediction and decision making. Big data, sophisticated information, and computational technology together are seen to be a viable way to improve operational effectiveness and make real-time monitoring possible throughout the food supply chain [74]. Moreover, real-time data visualization capabilities can be improved by using the intuitiveness of IoT networks and linked sensors to improve food supply chain packing, delivery, and sales [75]. These developments highlight how real-time processing technologies, which offer increased efficiency and decision-making skills, have the potential to completely transform the food sector. In Figure 10, the integration of IoT sensors with federated learning (FL) and AI models exemplifies how real-time data collection, aggregation, and processing can be utilized in quality control within the food supply chain. The system leverages environmental factors like temperature, humidity, and CO_2_ levels for monitoring agricultural products, enhancing both food quality assessment and overall supply chain efficiency. The real-time feedback provided by the system can facilitate immediate decision making to optimize production and minimize waste, paving the way for smarter, more sustainable agricultural practices.

#### 3.2.9. Scalability

Scalability is a significant challenge for AI in the food sector. The widespread adoption of AI technologies depends on the development of systems that can scale effectively to meet the demands of large-scale food production, distribution, and personalized services [76]. The success of implementing and maintaining healthy food retail initiatives is closely tied to the scalability of food computing systems. Factors influencing the scalability of healthy food retail interventions include the power of retailers in decision making, concerns about potential profit loss due to price reductions or product availability restrictions, and stakeholder dissatisfaction [77]. A systematic review of food computing highlights the key issues and future directions in the field, particularly the need for efficient collection and analysis of diverse food data from multiple sources to support functions such as food perception, recognition, retrieval, recommendation, and monitoring [1]. Additionally, a food traceability system based on blockchain and the IoT has been proposed to address food quality issues, though scalability and data availability challenges persist when using cloud computing technologies for food safety [78,79]. Addressing scalability concerns is essential to ensuring the effective implementation and long-term viability of AI-based solutions in food computing and related sectors, such as healthy food retailing. One example of addressing scalability in the food sector comes from the smart farming industry, where large-scale farms are using cloud-based solutions to manage data from numerous IoT sensors across multiple locations. For instance, a project integrated real-time data from thousands of soil moisture sensors and weather stations, allowing for large-scale crop management and resource optimization across vast agricultural areas [80].

#### 3.2.10. Interdisciplinary Collaboration

Interdisciplinary collaboration is essential to addressing the challenges of AI in food computing [8]. Developing comprehensive AI solutions for the food sector requires the integration of expertise from computer science, food science, and ethics [81]. Several research initiatives have focused on fostering interdisciplinary cooperation to tackle complex issues in the food sector. These initiatives emphasize critical areas such as food safety, environmental impact, health nutrition across different population groups, and the influence of the environment on sustainable food production and decision making [82]. Funding support for such initiatives often comes from both public and private sectors, reflecting the importance of collaborative efforts in advancing food system research. In addition, ethical considerations surrounding digital collaboration in the food sector have gained attention, with a focus on developing frameworks to guide the responsible use of technologies such as distributed ledger systems. Experts from AI, IoT, ethics, law, and risk management fields contribute to these efforts, underlining the need for a holistic approach to food safety and sustainability [83]. Moreover, interdisciplinary collaboration is recognized as key to overcoming rapid response challenges in the food supply chain. Researchers across diverse fields ranging from chemistry and microbiology to marketing and supply chain management are instrumental in fostering sustainable practices within the food sector. Emphasizing the need for clear conditions to support effective collaboration and information sharing is crucial to enhancing the impact of these efforts.

#### 3.2.11. Data Governance

Data governance is a critical concern for AI in the food sector. To ensure that AI is used responsibly and effectively in food computing, it is essential to establish clear data governance frameworks while addressing data security and privacy issues [84]. The emerging field of data governance in food computing aims to guarantee the ethical, secure, and high-quality use of data. For instance, the Global Partnership on AI’s Working Group on Governance discusses the significance, challenges, and best practices of data governance [85]. A checklist for practical data management decision making is also proposed [86]. Additionally, a collaborative architecture for blockchain-based enterprise data governance has been suggested to enhance data management in food computing [87]. These studies emphasize the importance of addressing ethical, security, and privacy considerations in data governance within the food sector.

#### 3.2.12. Standardization and Interoperability of Data

AI in the food sector faces challenges related to data standardization and interoperability [88]. To fully utilize AI in food computing, uniform data standards and interoperability across systems and data sources are crucial [89]. The European Food Safety Authority (EFSA) is working to digitalize EU food systems by standardizing data. The EFSA Advisory Group on Data aims to enhance food safety by connecting and integrating data streams from multiple sources, creating a centralized platform, and enabling interoperability through standardized terminology and high-quality metadata. One of the key tools under development is the FoodEx2 Smart Coding Application [70], which supports the EFSA’s multiannual work plan to improve data interchange relevant to food safety [90]. Data standardization and interoperability are essential to providing reliable solutions to current and future challenges in the food industry [85]. In supply chain traceability, for example, a major food retailer integrated IoT sensors from different manufacturers to monitor conditions such as temperature and humidity. By adopting an open-data protocol, the retailer achieved seamless integration across devices, enabling the real-time monitoring of food quality from farm to consumption [91,92].

The integration of artificial intelligence (AI) and the Internet of Things (IoT) has significantly transformed various industries, including the field of food computing. However, these advancements come with intricate and varied challenges, particularly in the food sector. To fully harness AI’s potential to transform food production, safety, and customer experience, these obstacles must be addressed. As the demand for smarter and more efficient food systems grows, there are critical areas where AI and the IoT can play pivotal roles.

### 3.3. The Role of Generative AI in Advancing Food Computing

Recent advancements in generative AI, such as OpenAI’s ChatGPT-4, offer exciting opportunities for transforming food computing by enabling more efficient data processing and personalized nutrition. These technologies can improve the analysis of large-scale IoT data, enhancing food quality monitoring, detecting defects, and enabling tailored nutritional recommendations [93]. However, despite these opportunities, there are notable challenges in applying generative AI to food systems, including data privacy concerns, ethical implications regarding algorithm transparency, and the technical feasibility of scaling AI models to handle complex, real-time food data. Moreover, combining blockchain technology with AI can enhance food data integrity and security, offering solutions for the transparency and trustworthiness required in food supply chains. As AI continues to evolve, it holds the potential to significantly improve food safety, personalized diets, and supply chain efficiency, but it also requires addressing these challenges to ensure its practical and ethical application in the food industry [94,95]. These advancements and challenges suggest the need for further exploration of the potential and limitations of generative AI in food computing, focusing on its applications, associated challenges, and how integrating blockchain technology can address some of these issues.

#### 3.3.1. Advancements in Generative AI for Food Data Processing and Personalization

Generative AI offers transformative capabilities in processing a large number of food-related data, especially in analyzing real-time data from IoT devices. AI models can identify patterns, detect anomalies in food products, and predict spoilage or contamination risks [94]. Additionally, generative AI can assist in crafting personalized nutrition plans by analyzing individual health data, preferences, and dietary requirements. This technology opens doors to providing tailored food recommendations, enhancing consumer health outcomes, and supporting the development of new food products [96].

#### 3.3.2. Challenges in Implementing Generative AI in Food Systems

Despite its potential, applying generative AI to food systems presents several challenges. Data privacy is a significant concern, especially when handling sensitive information such as consumer health data for personalized diets. Ethical issues around the transparency of AI models are also critical, as decisions made by “black-box” AI can lead to unintended biases or unfair outcomes. Furthermore, the scalability of AI models to handle large, real-time food data is a challenge that must be overcome to ensure AI’s effective integration into the food industry [97,98].

#### 3.3.3. Enhancing Food Supply Chain Transparency with Blockchain and AI

Combining blockchain technology with generative AI can greatly enhance the transparency, security, and integrity of food supply chains [99]. Blockchain’s ability to provide immutable records for tracking food origins, handling, and quality assurance ensures the transparency needed for consumer trust. When paired with AI, blockchain can also verify the accuracy of food data, providing further assurances against fraud and contamination [100]. Additionally, AI can optimize food supply chain management by analyzing trends, predicting shortages, and improving inventory control, leading to a more efficient and resilient food system [101].

## 4. Data Sources in Food Computing

Diverse data sources are used by food computing to stimulate innovation and knowledge in the field. These data sources form the basis for a wide range of applications, which include supply chain tracking, food safety, and customized nutrition recommendations. A strong framework for data integration is necessary to create dietary recommendations that are effective. The combination of restaurant databases, personal health sensors, and nutritional data is depicted in the diagram in Figure 11. Prior to running the data through an Expert Health Knowledge Engine, this system uses a data quality filter to guarantee accuracy. Better health management is made possible by this integrated method, which generates customized dietary recommendations for consumers.

### 4.1. Types of Data

Images: In food computing, visual data are essential, particularly to applications like quality evaluation, calorie estimation, and food recognition [102]. Computer vision models are trained with high-resolution photographs of food products in order to effectively detect and classify various food categories. Publicly available datasets include the following:–Food-101: Frequently used for training image recognition models, it has 101,000 photos categorized into 101 distinct food types [103]. João Louro et al. used the Food-101 dataset to analyze food ingredient recognition by leveraging a CNN (ResNet-50) combined with fine-tuning and transfer learning techniques. Their approach aimed to enhance food identification accuracy while addressing the dataset’s limitations, such as the overrepresentation of Asian foods [104]. Prakhar Tripathi explored transfer learning in deep neural networks by using the Food-101 dataset to build a food classifier, emphasizing improved training efficiency and accuracy [105]. Ignazio Gallo et al. used the UPMC Food-101 variant to investigate multimodal classification by fusing image and text data through BERT and CNNs, achieving superior performance with an early fusion stacking approach [106].–UECFOOD-256: Absolutely ideal for food classification, this image collection includes pictures from 256 food categories, with a primary focus on Japanese food [107]. Berker Arslan et al. focused on food classification using the UECFood-256 dataset, where they explored various deep learning methods for fine-grained food recognition. Their work achieved a State-of-the-Art (SOA) accuracy of 90.02% on the UEC Food-100 database, which has been extended to the UECFood-256 dataset. They emphasized the use of ensemble methods, particularly combining ResNeXt and DenseNet models, and proposed the first averaged-trial comparison, setting a new benchmark for food classification [108]. Elena Battini Sönmez et al. addressed food detection by introducing the Segmented UEC Food-100 dataset, which includes segmentation masks. Although their primary focus was on UEC Food-100, the methods are applicable to the UECFood-256 dataset for multi-food item detection. They compared different segmentation approaches, achieving an mIoU of 64.63% with YOLAC and an mAP of 68.83% with YOLACT in instance segmentation, offering significant contributions to food detection and classification on these challenging datasets [109].–Food-5K: A dataset created for diet-tracking and smart kitchen applications that detects food in real-world settings [110]. Shuang Liang and Yu Gu proposed a multi-stage CNN framework for food recognition, incorporating innovative modules such as a boundary-aware module (BAM) for detecting boundary regions, deformable ROI pooling (DRP) for spatial feature refinement, and a transformer encoder for capturing global contextual relationships. Their framework achieved SOA performance with a Top-1 accuracy of 99.80% on the Food-5K dataset, significantly outperforming existing methods. This framework demonstrates great potential for applications in automated meal tracking and personalized nutrition planning, providing robust solutions for real-world dietary management [111].Nutritional information: Food product macronutrients (proteins, carbs, and fats), micronutrients (vitamins and minerals), and other dietary components are all included in nutritional data [112]. Personalized nutrition, health tracking, and diet planning applications all require this information to be developed. Publicly available datasets include the following:–The Yummly dataset helps with nutrition analysis and recommendation systems by offering thorough nutritional data for a wide variety of recipes [113]. Thomas Theodoridis et al. proposed a cross-modal variational framework for food image analysis, focusing on ingredient recognition. The framework processes information from both image and text modalities by using two variational encoder–decoder branches, while a variational mapper aligns the distributions of the branches. Experimental results on the Yummly-28K dataset demonstrated the framework’s superiority over similar methods, achieving better performance than current SOA approaches on the large-scale Recipe1M dataset [114]. Viswanath C. et al. applied an InceptionV3-based CNN model for food image classification. Their approach used convolution layers capable of generating their own kernels to convolve with the input layer, along with a Max-Pooling function for feature extraction. By combining multiple layers and concatenating the outputs, they achieved a notable accuracy of 92.89% on both the Yummly dataset and their own dataset, demonstrating the effectiveness of their CNN-based model in food recognition [115].–Recipe1M provides nutritional information in addition to its one million recipes, which is helpful for both dish suggestions and nutrition research [116]. RecipeGM by Anja Reusch et al. introduces a hierarchical recipe generation model using CNNN with self-attention mechanisms for generating recipes based on a given set of ingredients. This model outperforms RecipeGPT in some cases, and it addresses evaluation challenges in recipe generation [117]. FMI (Fine-grained Modalities Interaction for Cross-Modal Recipe Retrieval) by Fan Zhao et al. improves cross-modal recipe retrieval by leveraging a hierarchical recipe transformer, the CCMRE module for enhancing recipe components, and the TCVE module to enrich the visual encoder. Their approach outperforms the SOA on the Recipe1M dataset, with significant improvements in retrieval accuracy [118].Recipes: Ingredient lists, preparation techniques, cooking durations, and portion sizes are all included in recipe data. Recipe analysis facilitates dietary restriction modification, knowledge of culinary patterns, and the creation of meal suggestions [119].Food Logs: Food logs are records that track a person’s daily food consumption. These logs provide valuable insights into dietary choices, behaviors, and intake, which are essential to behavior analysis and health-related applications [120].

### 4.2. Data Collection Methods

Crowdsourcing: Crowdsourcing is the process of collecting data from a big number of individuals, usually via internet sites. This method allows for the aggregation of diverse data types from a wide audience, which is particularly valuable in food computing applications, where varied perspectives and inputs are crucial. This approach works well for gathering a wide range of information, including recipe books, food logs, and user-submitted photos of food. Annotating data with crowdsourcing improves the quality and usefulness of datasets [121,122]. An ecosystem for food delivery that links food producers, consumers, and crowdsourced riders is depicted in the following diagram in Figure 12. Customers use a computer or smartphone to submit orders, which are subsequently accepted by food providers, to start the process. The crowdsourced riders are coordinated by the cloud-based system, which also makes order management easier. After logging in, riders are assigned delivery routes. By using shared motorbikes for transportation, this integrated approach guarantees effective food delivery from providers to clients.Sensors: Smart kitchen appliances, wearable technology, and smartphones all have sensors built in to gather data on food preparation and consumption in real time [123]. These sensors play a crucial role in food computing by providing precise, real-time data, enabling the dynamic and accurate monitoring of food-related activities. For instance, temperature sensors keep track of cooking conditions and can estimate portion amounts, while weight sensors can evaluate food safety and quality management [124].Internet of Things (IoT): Data on food can be continuously collected and transmitted through IoT devices [123]. The IoT represents a fundamental technological framework for enabling the seamless flow of information across various devices and systems, thereby enhancing the capabilities of food computing. For example, smart refrigerators may monitor stock levels and expiration dates, and kitchen equipment that is connected can give specific consumption trends. The IoT makes it possible to seamlessly integrate different data sources, which improves the accuracy and comprehensiveness of applications related to food computing [125].

## 5. The Convergence of AI and the IoT in the Food Sector

The combination of AI and the IoT is transforming food computing in several ways, including supply chain optimization, quality control, food safety, sustainable practices, and predictive maintenance of equipment. Some of the key applications and benefits are reported below.

### 5.1. Supply Chain Optimization

It is becoming more widely acknowledged that combining AI and the IoT with food computing is a crucial strategy for improving supply chain management in the food sector. Aspects of this convergence have been examined in a number of recent research studies, including supply chain optimization using ML models, AI-based yield prediction systems, and drones for crop monitoring and data gathering. These technologies have shown a great deal of promise for increasing productivity, cutting waste, and guaranteeing food safety throughout the food supply chain [126]. Everloo et al.’s Leveraging AI in Food Product Development explores how AI is used in food innovation and how the food sector is affected by it [127]. The paper emphasizes how AI may be used to improve supply chain processes, spur product innovation, and provide wholesome, sustainable food options that meet customer demand. Food traceability was suggested by Sharma et al. [128], who suggested a blockchain-based and IoT-based system. The suggested work is transparent and self organized, with the goal of improving the food supply chain’s efficiency. To address consumer demand for high-quality products and assure food safety, AI-based quality control methods have been applied in addition to these studies. These systems anticipate and stop quality problems in food products by analyzing data gathered from several sources, including sensors and human experts, by using ML algorithms [129]. Furthermore, to ensure the successful adoption and acceptance of these innovations, it is imperative to comprehend customer perspectives on supply chain optimization through the use of AI and IoT technology. Consumer expectations regarding product availability, quality, and sustainability can be learned a great deal from surveys or research findings on consumer attitudes toward AI and the IoT in supply chain management [130].

### 5.2. Quality Control and Food Safety

There has been increasing interest in the convergence of AI and the IoT in the field of the food sector, particularly in quality control and food safety. Food safety and quality control procedures could become much more accurate and efficient with the inclusion of AI and IoT technology [131]. AI algorithms, for instance, may evaluate information from IoT sensors to identify possible risks to food safety, and IoT devices can track the conditions of food storage and transit in real time [73]. In the context of fresh food logistics, where there is an increasing need for food products of a higher caliber, solutions integrating distributed ledger technologies, digital twins, the IoT, and AI [63] are being proposed, and this convergence is also being investigated. These developments have the potential to completely transform the food sector, improving its efficiency, safety, and ability to satisfy consumers’ growing need for premium food items. A key factor in the adoption of AI and IoT technologies in food computing, especially in quality control and food safety, is an understanding of consumer trust and acceptance. Understanding customer trust and concerns can help steer the development of better consumer friendly innovations and communication tactics, according to research findings or surveys on consumer perceptions and the adoption of AI and the IoT in food computing [132]. Figure 13 illustrates a sustainable food supply chain that begins with the production of crops at the farm, followed by harvesting, processing, and transportation to consumers. After consumption, the waste management process is depicted, showing recycling, composting, or disposal. The key feature of this system is the circular flow, where waste or by-products are returned to the farm, either as compost or fertilizers, promoting sustainability and closing the loop in the food production process.

### 5.3. Sustainable Practices

The potential for sustainable practices in food computing through the convergence of AI and the IoT is highlighted in recent publications. Bourechak et al. in [133] delve in the possible applications of edge computing and AI in a range of fields, such as smart agriculture and food recognition. Vilas et al. [63] make recommendations for using distributed ledger technology, digital twins, the IoT, and AI to produce food products that are fresher and of higher quality. In order to manage complex systems in smart cities, blockchain is increasingly being combined in novel ways [63,134]. Sharma et al., in [135], suggest a strategy to maximize precision agricultural techniques that embrace the confluence of AI, ML, and the IoT. The promise of AI and IoT convergence in the food sector for sustainable practices, such as supply chain management, precision agriculture, and food safety, is illustrated in these studies. Figure 14 demonstrates a four-layer system designed for optimizing apple harvesting under ideal environmental conditions. The Sensing Layer collects real-time data by using environmental sensors that monitor factors such as temperature, humidity, and air quality. The Network Layer facilitates the transmission of these data through IoT protocols for seamless communication. In the Service Layer, the data are processed and analyzed to assess the ripeness of apples and the environmental conditions for optimal harvesting. The Application Layer features an automated harvesting machine that uses this information to selectively harvest apples that are ripe and in the perfect condition for picking, ensuring precision and efficiency in the harvest process. We may investigate how developments in AI and the IoT can support personal nutrition, encouraging healthy diets and reducing problems like food allergies and sensitivities, by incorporating viewpoints from nutritional research [136]. Environmental science directs the development of technologies which reduce resource consumption and environmental impact by providing crucial insights into the ecological footprint of food production and delivery. Furthermore, social scientific understanding illuminates the socioeconomic dynamics of food systems, encompassing concerns about cultural preferences, equality, and food access [137]. By combining these various viewpoints, we may create comprehensive solutions that not only tackle technological difficulties but also social demands and environmental issues, thus encouraging a more sustainable food ecosystem [138].

### 5.4. Predictive Maintenance of Equipment

The possibility for predictive equipment maintenance through the confluence of AI and the IoT in the food sector has been brought to light by recent studies. Raza et al. [139] show the importance of AI and the IoT in creating predictive maintenance systems which may identify equipment problems before they happen and minimize downtime while boosting productivity. An additional study by Addanki et al. [140] demonstrated how AI and the IoT in the food sector might enhance food safety and quality by tracking and regulating many factors like temperature, humidity, and air quality. A paper by Singh et al. [141] explored how blockchain and the IoT may help create a transparent and safe food supply system, which would help build sustainable smart cities. Furthermore, AlZubi et al., in [142], emphasized how AI and the IoT might enhance the sustainability and efficiency of food production and delivery systems. These studies show how the integration of the IoT and AI in food computing has the potential to completely transform the food sector by providing new avenues for sustainability, efficiency, and safety improvements.

### 5.5. Ethical Frameworks for Food Computing

In order to ensure an ethical implementation of AI and the IoT in the food sector, several ethical frameworks and principles exist. These need to cover important factors including access, equity, and the effect on underserved groups. The “Ethical AI for Food” norms which were developed in 2023 by the Global Partnership on AI (GPAI) represent one such framework. Principles for the ethical application of AI in the food sector are outlined in this framework. These include environmental sustainability, fairness and equality, respect for human rights, and accountability and transparency [143]. Furthermore, the standards place a strong emphasis on protecting individual privacy and data rights, guaranteeing algorithmic fairness, and placing high priority on human wellness [144,145]. All parties involved in food computing technologies, i.e., developers, legislators, and sector executives, can cooperate to responsibly integrate AI and the IoT by embracing these ethical standards. This will lessen the chance of things like unfair access to technology, biased decision making, and the marginalization of people that are already vulnerable. An example of ethical data handling in the food sector can be seen in the implementation of GDPR-compliant practices in food delivery services. For instance, a popular food delivery service has integrated GDPR by anonymizing user data, including dietary preferences and order histories, ensuring that explicit consent is obtained before collecting any personal information. The service also offers users the option to opt out at any time, and all personal data are securely stored and used solely to enhance user experience. This approach not only guarantees GDPR compliance but also builds consumer trust, demonstrating how ethical frameworks can be practically applied to protect privacy in IoT-enabled food systems [146,147].

## 6. Future Directions of AI and IoT in Food Computing

The future of AI and the IoT in food computing promises transformative advancements in efficiency, sustainability, and personalization. One major trend is the integration of AI-driven predictive analytics with IoT sensors to optimize food production and supply chain management. These technologies enable the real-time monitoring of soil health, weather conditions, and crop growth, allowing farmers to enhance yields while minimizing resource waste. Studies show that IoT-based smart farming systems, when combined with AI, can reduce agricultural production costs by up to 20% [148]. Beyond production, AI and the IoT will play a crucial role in food safety and traceability. The integration of blockchain with IoT devices is expected to enhance transparency across the supply chain, ensuring the precise tracking of food products from farm to table. This will strengthen compliance with safety regulations and boost consumer trust [149]. However, challenges such as high computational costs and data standardization issues may slow widespread adoption. In food consumption and storage, smart appliances such as IoT-enabled ovens and refrigerators will become more prevalent. These devices can be remotely controlled and personalized based on user preferences, improving convenience and product quality [150]. Another promising direction is AI-powered personalized nutrition, where dietary recommendations are tailored based on an individual’s health data and lifestyle habits. This aligns with the rising consumer demand for health-conscious and customized food solutions [151]. However, privacy concerns related to the collection and use of personal health data must be carefully addressed. The concept of digital twins, virtual replicas of physical systems, is also gaining traction in food production. These simulations will allow manufacturers to optimize production processes, reduce waste, and improve sustainability [152]. Furthermore, the adoption of edge computing in IoT devices is expected to enable faster data processing at the source, reducing latency in decision making for critical applications such as food safety monitoring and logistics optimization [153]. While these advancements offer significant potential, several barriers must be overcome, including data security risks, regulatory compliance issues, and the high costs associated with large-scale implementation. Addressing these challenges will be crucial to the successful deployment of AI and the IoT in the food sector. As we look ahead, several key areas emerge where AI and the IoT are poised to have a significant impact on the future of food computing.

### 6.1. AI-Driven Predictive Analytics in Smart Agriculture

The integration of AI with IoT-enabled precision agriculture will allow for the real-time forecasting of crop diseases, yield prediction, and automated resource management [154]. Future developments should focus on enhancing model accuracy, interpretability, and adaptability to different environmental conditions. Lightweight AI models are needed to reduce computational burdens on IoT devices operating in remote farming regions. Additionally, integrating multimodal data sources such as satellite imagery, soil sensors, and climate data will improve decision making and resource optimization [155].

### 6.2. Blockchain-Enabled Food Safety and Traceability

Combining blockchain with IoT tracking systems can ensure food safety and improve supply chain transparency. Research should explore ways to optimize blockchain scalability, enhance interoperability with existing standards, and develop energy-efficient consensus mechanisms [156]. Additionally, AI-powered anomaly detection can identify potential contamination risks in the supply chain before they impact consumers. Future advancements should also focus on automating regulatory compliance by using AI to monitor and verify food safety standards in real time [157].

### 6.3. Smart Appliances and Personalized Nutrition

AI-powered IoT-enabled refrigerators, ovens, and dietary assistants will provide personalized recommendations based on health data and dietary habits [158]. Future advancements should emphasize privacy-preserving AI techniques to address concerns related to user data security. Additionally, integration with wearable health monitors can provide a holistic approach to nutrition management. AI-driven meal planning and automated grocery shopping assistants could further enhance convenience and promote healthier eating habits [159].

### 6.4. Digital Twins in Food Production

Digital twin technology will revolutionize food production by enabling real-time simulations of food manufacturing and processing workflows [160]. AI-driven digital twins can predict inefficiencies, reduce waste, and optimize resource utilization. Future research should focus on adaptive learning mechanisms that allow digital twins to continuously update their models based on real-time IoT data. By incorporating ML and sensor-driven feedback loops, food manufacturers can enhance production efficiency, detect potential faults in processing lines, and minimize energy consumption [161].

### 6.5. Edge Computing for Real-Time Food Monitoring

Edge computing will become essential in food computing by reducing latency and enabling real-time decision making for applications such as food spoilage detection, logistics optimization, and precision irrigation. The key challenges include developing energy-efficient AI models capable of running on low-power edge devices and enhancing cybersecurity measures to protect IoT networks from cyber threats [162,163]. Future efforts should explore FL techniques to enable collaborative AI training across distributed edge devices while preserving data privacy.

While these advancements offer significant potential, several barriers must be overcome, including data security risks, regulatory compliance issues, and high costs associated with large-scale implementation. Future research should emphasize the development of standardized data-sharing protocols, privacy-preserving AI techniques, and industry-wide collaboration to ensure sustainable deployment.

## 7. Discussion

In recent years, the intersection of AI and the IoT has catalyzed transformative advancements in the food sector, addressing complex challenges across the supply chain, quality control, and sustainability. Table 2 provides a comprehensive overview of key findings, highlighting the uses, challenges, and potential solutions identified in the literature. This table incorporates a critical analysis of issues such as data quality, regulatory compliance, scalability, and interdisciplinary collaboration, shedding light on the limitations of existing approaches. Addressing challenges like ethical concerns, environmental impact, and real-time processing capabilities is paramount to unlocking the full potential of AI and the IoT in food computing. Moreover, the studies reviewed emphasize the importance of robust data governance frameworks, globally harmonized standards, and collaborative efforts among diverse disciplines. Future research must prioritize the development of scalable AI architectures, energy-efficient algorithms, and culturally adaptive personalization models. Additionally, advancing privacy-preserving techniques and fostering cross-disciplinary approaches will be essential to overcoming current barriers and maximizing the impact of AI and IoT innovations in transforming global food systems.

Despite the progress outlined in Table 2, several unresolved challenges persist, limiting the full-scale implementation of AI and the IoT in food computing. A key issue is the fragmented nature of data collection and data-sharing mechanisms, which hinders the development of reliable, globally representative datasets. Additionally, current AI-driven solutions often struggle with scalability, particularly in resource-constrained environments where computational efficiency and energy consumption are critical concerns. Ethical and privacy-related challenges remain a pressing issue, as data governance frameworks vary across regions, creating inconsistencies in compliance and enforcement. Future research should focus on the development of standardized protocols for cross-border regulatory alignment, ensuring that AI applications adhere to global food safety and privacy regulations. Moreover, enhancing the adaptability of AI models to diverse cultural and dietary preferences remains an open research direction, requiring interdisciplinary collaboration among food scientists, AI researchers, and policymakers. Addressing real-time processing bottlenecks through edge computing and FL will be essential to enabling rapid decision making in food quality assessment and supply chain management. By tackling these challenges, future advancements can drive more efficient, ethical, and scalable AI-powered solutions in the food sector.

## 8. Conclusions

This review paper has explored the evolving intersection of AI and the IoT within the food computing sector. Through a comprehensive analysis, we have demonstrated how AI is revolutionizing key areas of the food supply chain, including supply chain optimization, quality control, and the promotion of sustainable practices. However, the technological advancements discussed here must be balanced with a nuanced understanding of the policy and regulatory frameworks required to ensure their responsible and equitable application. Governments, industry leaders, and other stakeholders must collaborate to establish robust regulations that prioritize ethical considerations, strong data governance, and privacy protection, ensuring that AI technologies are deployed in a manner that is both responsible and beneficial to all parties involved. We also examined the diverse range of data used in food computing, such as nutritional data, food logs, and images, alongside crucial data collection methods like IoT devices and sensors. These data sources are pivotal to driving the ongoing development of AI and IoT technologies in food systems, fostering progress in areas such as food safety, sustainability, and personalized nutrition. As these technologies continue to mature, future research must focus on resolving challenges related to data interoperability and scalability, as well as addressing the ethical concerns surrounding AI in the food sector. It will be essential for researchers to collaborate across disciplines to ensure that AI can be integrated in a way that maximizes its benefits while minimizing its risks. Looking ahead, addressing the issues of real-time processing, scalability, and environmental sustainability will be critical. The seamless integration of AI and the IoT in food systems presents the opportunity to improve efficiency, reduce waste, and enhance food safety. By embracing innovative approaches and continuing to prioritize ethical considerations, future research can pave the way for a more sustainable, secure, and effective food sector.

## Figures and Tables

**Figure 1 sensors-25-02147-f001:**
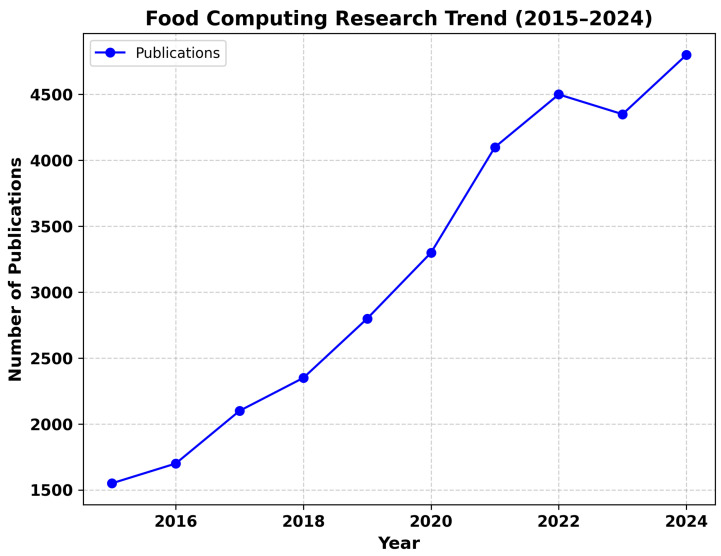
Yearly distribution of research publications in food computing indexed in Web of Science from 2015 to 2024.

**Figure 2 sensors-25-02147-f002:**
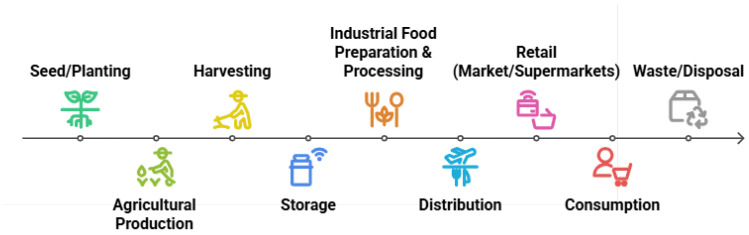
Food supply chain process: stages from seed to consumer.

**Figure 3 sensors-25-02147-f003:**
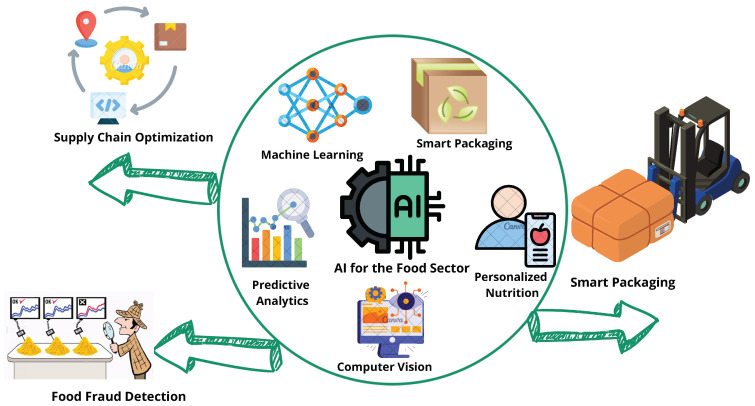
AI in food applications.

**Figure 4 sensors-25-02147-f004:**
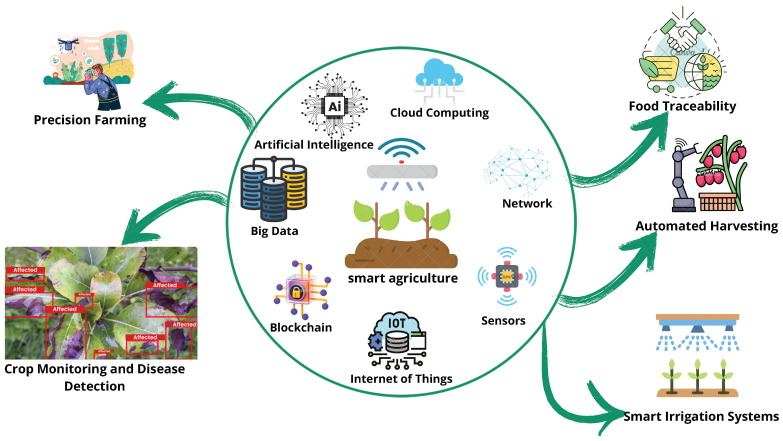
IoT in food applications.

**Figure 5 sensors-25-02147-f005:**
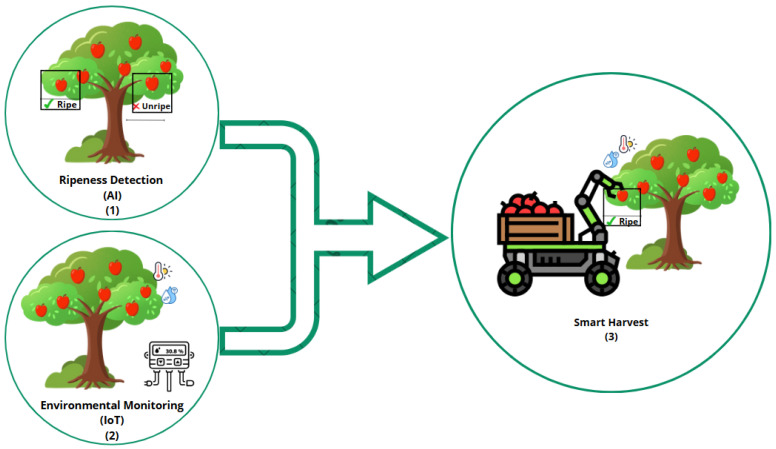
The integration of AI for apple ripeness detection and the IoT for environmental monitoring to ensure optimal harvesting conditions: (1) AI-based detection is used to assess the ripeness of apples, while (2) IoT sensors monitor environmental conditions like temperature and humidity. By merging these two technologies, (3) we create a system that ensures apples are harvested at their optimal ripeness and under ideal environmental conditions, enhancing both quality and efficiency in the harvesting process.

**Figure 6 sensors-25-02147-f006:**
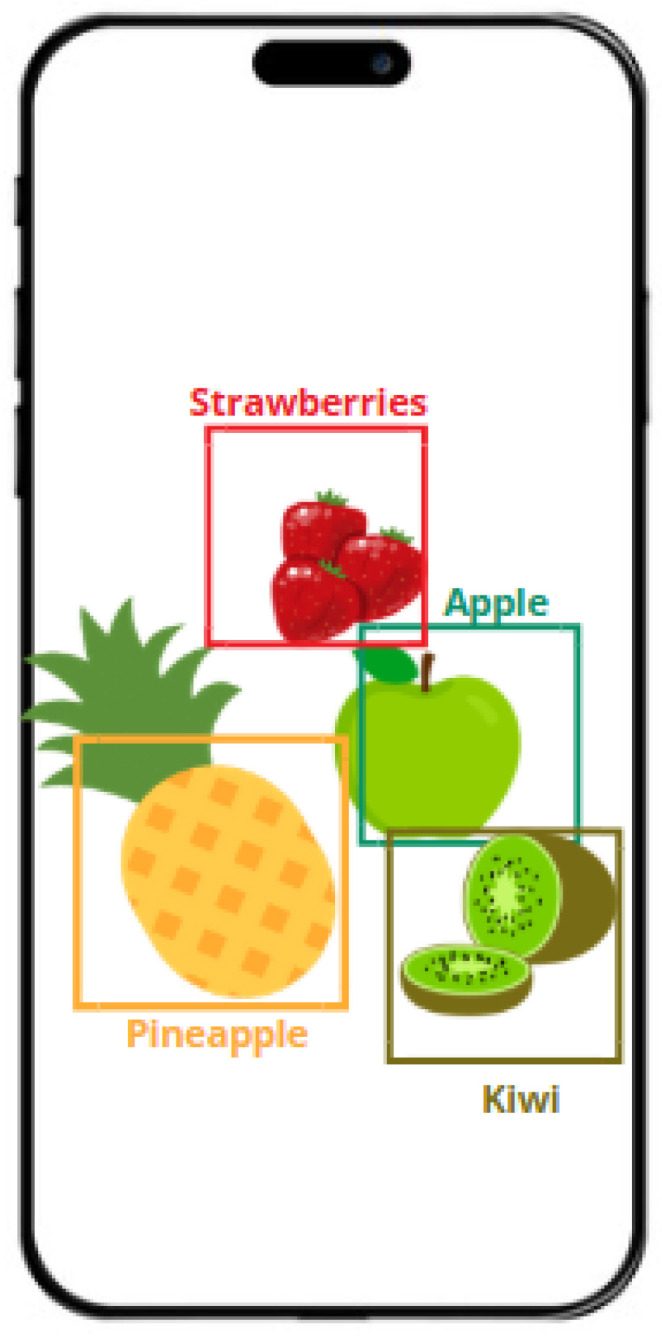
Fruit recognition: kiwi, pineapple, apple, and strawberries.

**Figure 7 sensors-25-02147-f007:**
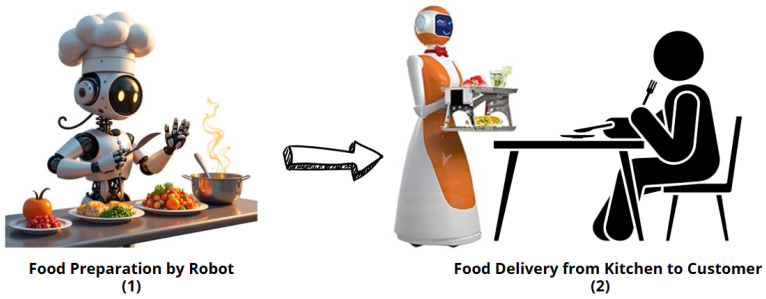
Intelligent restaurant system: (1) A robot prepares food in the kitchen, using automated systems to cook and assemble dishes quickly and accurately. (2) Another robot delivers the prepared food directly to the client’s table, improving service speed and creating a more efficient dining experience.

**Figure 8 sensors-25-02147-f008:**
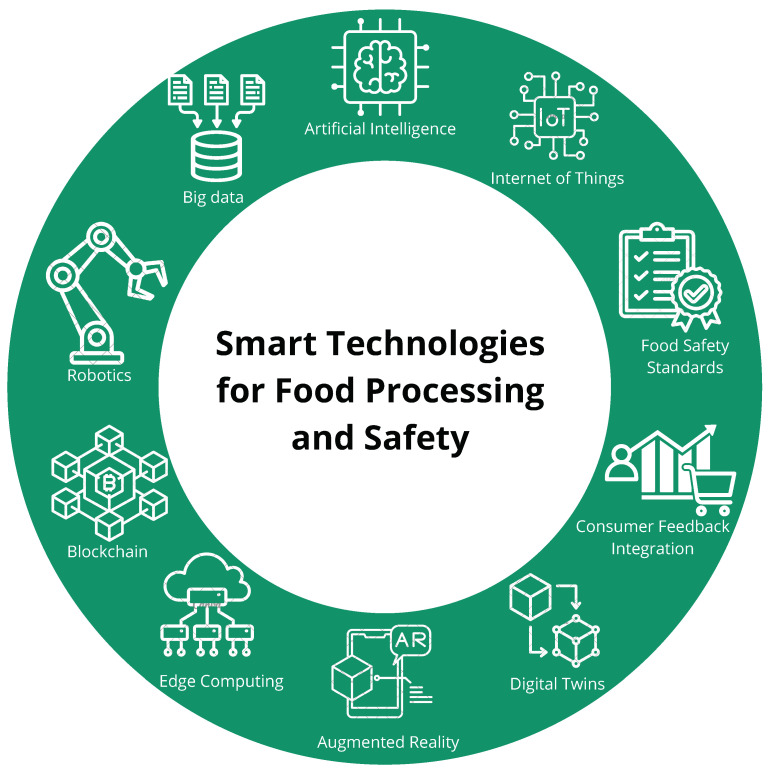
Important components of Industry 4.0 technology for food processing.

**Figure 9 sensors-25-02147-f009:**
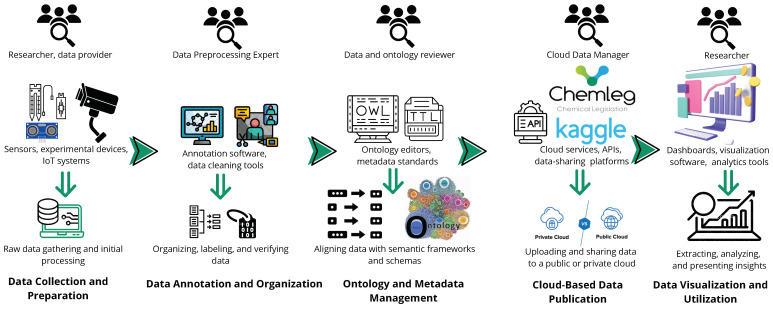
End-to-end data workflow in research ecosystems: The figure shows the process of managing research data. It starts with collecting and processing data, followed by organizing, annotating, and reviewing them to ensure quality. Ontology management is used to structure and connect the data. The data are then stored in the cloud and shared in a community warehouse. Finally, they can be extracted, visualized, and used by researchers and professionals for their work.

**Figure 10 sensors-25-02147-f010:**
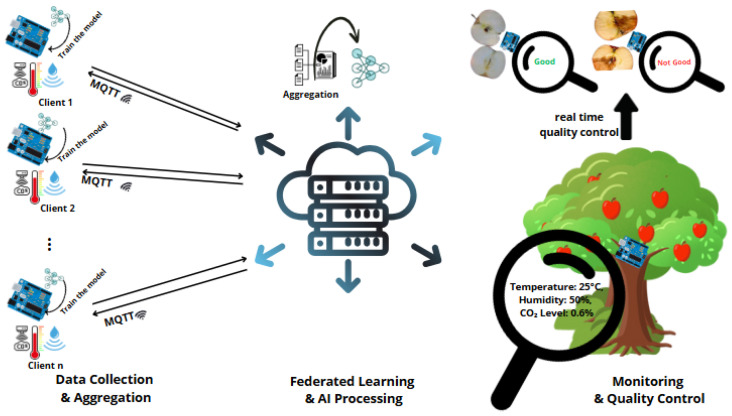
AI-driven monitoring and quality control system with FL: an AI-enhanced system for monitoring and quality control in the agri-food sector. The system integrates data collection and aggregation from multiple IoT clients (Client 1 to Client n) by using MQTT communication. The collected data are sent to the cloud, where FL and AI processing are employed to analyze environmental factors like temperature, humidity, and CO2 levels. The results of this processing are used for real-time quality control, which assesses product quality (e.g., apples and other fruits) based on collected environmental data. The system optimizes agricultural practices by ensuring the quality of the produce through continuous monitoring and AI-powered quality checks.

**Figure 11 sensors-25-02147-f011:**
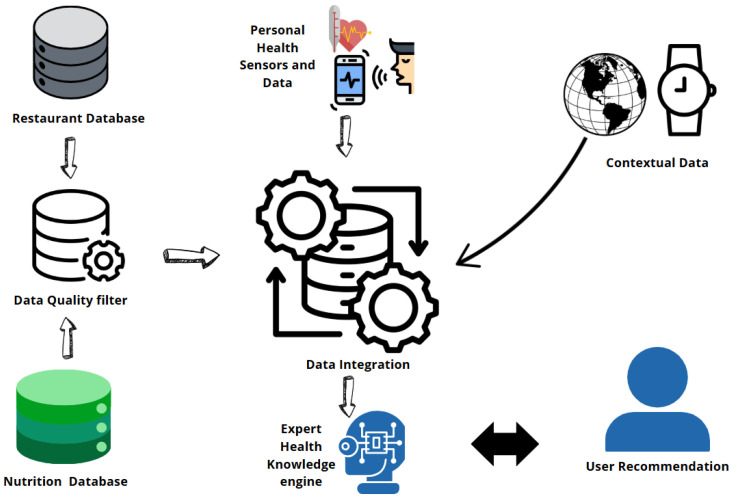
Integration framework for personalized nutrition: merging health data and nutrition databases to provide tailored dietary recommendations.

**Figure 12 sensors-25-02147-f012:**
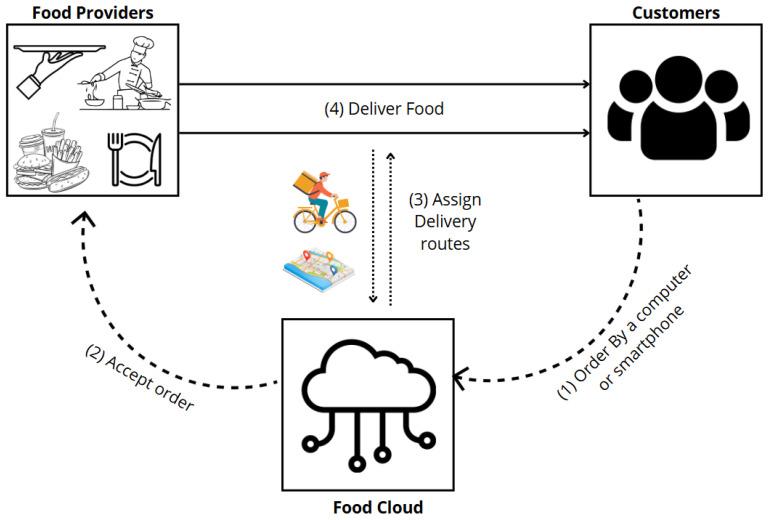
Food delivery ecosystem: The figure illustrates the food delivery process in four steps: (1) Customers place their order via a computer or smartphone. (2) Food providers accept the order and begin preparing the food. (3) Delivery routes are assigned to ensure efficient delivery. (4) The food is delivered to the customer. This process highlights the integration of digital technologies to ensure a seamless and efficient food delivery service.

**Figure 13 sensors-25-02147-f013:**
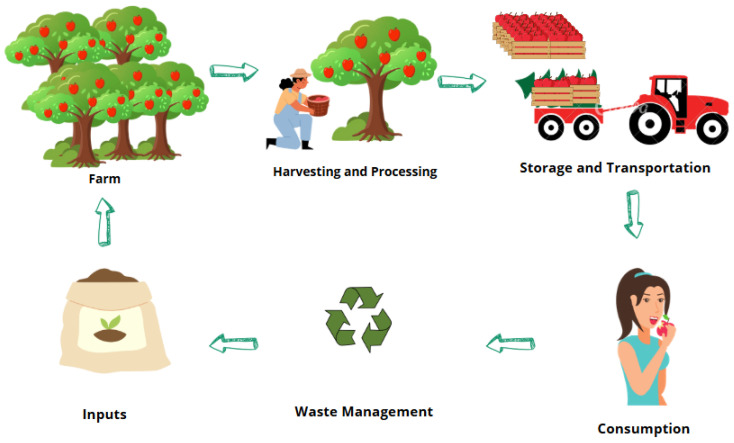
Food supply chain: from farm to consumption with circular waste flow.

**Figure 14 sensors-25-02147-f014:**
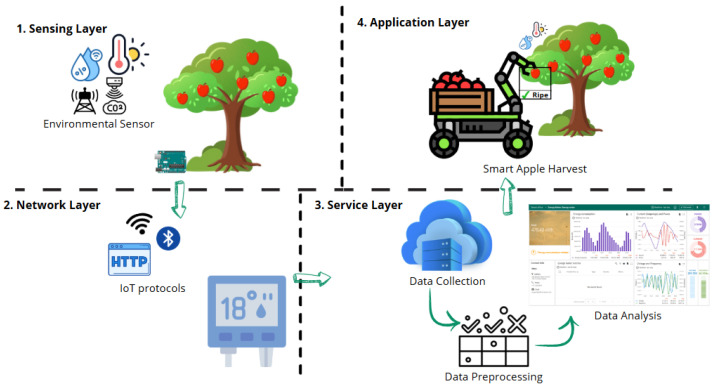
Four-layer system for real-time apple harvesting based on environmental and ripeness conditions.

**Table 1 sensors-25-02147-t001:** Comparison between food and food computing: The table shows the differences between food and food computing. While food is mainly about physical properties and nourishment, food computing uses technology like AI and the IoT to improve food quality, safety, and traceability. Food computing brings new ways to manage and enhance food processes through technology.

Aspect	Food	Food Computing
Involves physical properties	✓	✗
Uses computational methods	✗	✓
Enhances food quality and safety	✗	✓
Involves AI or IoT	✗	✓
Focuses on nourishment	✓	✗
Focuses on technology applications	✗	✓

**Table 2 sensors-25-02147-t002:** Summary of cited papers with future research directions and critical analysis.

Research Area	Findings	Future Research Directions	Critical Analysis
Data quality and availability [44,45,46,47,48]	High-quality data for ML technologies in the food sector. Issues remain in data collection, sharing, and application.	Investigate data-sharing protocols and scalable solutions for real-time data collection across the food supply chain.	Existing studies lack emphasis on data quality in resource-constrained environments, limiting their global applicability.
Complexity of food attributes [37,49,50]	Understanding the relationship between food complexity and consumer behavior. The complexity of food attributes affects AI algorithms.	Explore AI-driven models that integrate consumer behavior data to predict food preferences and health outcomes.	Current approaches fail to address the subjective nature of food complexity, which varies across cultural and demographic groups.
Personalization and cultural differences [50,51,52]	Personalization and cultural differences present challenges for AI in the food sector. Understanding cross-cultural food preferences is essential to personalized food recommendations.	Conduct research on AI-based solutions for personalized nutrition across diverse cultural contexts.	Datasets used for personalization often lack representation from non-Western dietary habits, leading to biased models.
Ethical and privacy concerns [51,53,54,55,56,57]	Ethical and privacy concerns pose challenges for AI adoption in the food sector. Regulation and oversight are needed to ensure responsible AI deployment.	Explore best practices for data anonymization in AI food systems, and investigate regulations like GDPR in international contexts.	Despite existing frameworks, enforcement of ethical practices and compliance with privacy laws is inconsistent, especially across borders.
Environmental impact [58,59,60,61,62]	AI applications in food computing have the potential to reduce environmental impact. Challenges remain in minimizing energy consumption and waste production.	Research the development of energy-efficient algorithms and sustainable AI solutions for reducing food waste.	Most studies overlook the lifecycle emissions of AI systems, especially in large-scale industrial applications.
Regulatory compliance [63,64,65,66,67]	Regulatory compliance for AI adoption in the food sector. Issues include aligning AI technologies with legal requirements and ensuring data security.	Investigate frameworks for cross-border regulatory compliance and standardization in AI-driven food systems.	Current research lacks actionable guidelines for navigating conflicting regulations across regions.
Integration with existing systems [68,69,70,71,72]	Integration with existing systems for AI adoption in the food sector. Issues include interoperability and compatibility.	Study the integration of emerging AI technologies with legacy food systems and the role of the IoT in enhancing system interoperability.	Studies often fail to propose scalable solutions for integrating AI with legacy systems without major infrastructure overhauls.
Real-time processing challenges [73,74,75]	Real-time processing ensures food safety and quality. Challenges include data latency and processing speed.	Explore the use of edge computing to enable faster, real-time data processing at the source of food production.	Current solutions for real-time processing often prioritize speed over accuracy, leading to trade-offs in decision quality.
Scalability [1,76,77,78,79,80]	Scalability remains a challenge for AI solutions in food computing, particularly with the increasing volume and variety of data. Proposed solutions involve parallel computing, FL, and containerization of AI workloads.	Develop frameworks for scaling AI models across distributed networks in food systems, focusing on FL for privacy preservation.	Scalability challenges are compounded by the lack of robust frameworks for integrating heterogeneous datasets in food computing.

## Data Availability

Data are contained within the article.

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
