# Peer review of "AI-Enabled IoT for Food Computing: Challenges, Opportunities, and Future Directions"

_sensors, 2025, doi:10.3390/s25072147_

Round 1
Reviewer 1 Report
Comments and Suggestions for Authors
The article focuses on applications of artificial intelligence such as speech recognition for food analysis, computer vision for food identification and quality evaluation, and predictive modeling for nutritional advice. In order to enable traceability and customized solutions, the integration of AI technologies depends on a variety of data sources, including user-generated logs, nutritional databases, and photographs. The study investigates options such as blockchain for transparent traceability and federated learning for safe, on-device data processing to address these issues. There is also discussion of new topics like AI-IoT integration-powered green procedures and edge AI for real-time analytics. This review provides practical suggestions for advancing the food industry using creative and moral technological platforms.
This paper deals with the rising demands on food that turns out to no longer be handled using conventional food production and distribution techniques. Hence, integrating contemporary technology like artificial intelligence (AI) and the Internet of Things (IoT) has become a viable way to address these issues. The constantly growing field of food computing uses data analytics, IoT, and AI to improve food systems at every stage, from production and processing to distribution and consumption. Food systems can be made more effective, sustainable, and adaptable to changing circumstances by utilizing AI's capacity to analyze massive datasets and IoT's real-time monitoring capabilities. When combined, these technologies could help solve urgent global problems like waste reduction, tailored nutrition, and food insecurity.
The present article examines how the food industry is changing as a result of AI and IoT integration. The authors look at how they are used for environmentally friendly procedures, supply chain optimization, and food quality monitoring. The authors also identify the difficulties in incorporating these technologies into the food industry, including scale constraints, privacy concerns, and data heterogeneity, and offers suggestions for overcoming these obstacles. In addition, the authors assess the current difficulties in food computing critically with the goal of offering a thorough examination of the approaches and tactics employed in recent studies. By concentrating on these issues, this paper hopes to direct future scholars and practitioners, assisting them in concentrating on the areas that require more investigation and advancement. The approach adopted by the authors guarantees a careful review of pertinent scholarly works and offers a solid basis for incorporating knowledge about the dynamic interplay between AI and IoT in the context of food computing.
The steps followed by the authors are clear and the paper does not require any further improvements in terms of methodology.
The conclusion is well written and summarizes the work approach
The list of references is adequate and completely summarizes the state of art findings on the topic of the paper.
Author Response
Thank you for your positive feedback and thoughtful summary of our paper. We appreciate your recognition of the methodology, conclusion, and comprehensive references. We are glad that the paper's approach and insights were well received. Your summary effectively captures the core aspects of our work, particularly the integration of AI, IoT, and data-driven methodologies in Food Computing. The emphasis on computer vision, natural language processing, and predictive modeling for food-related applications aligns well with our discussion on AI advancements. Similarly, we appreciate your acknowledgment of the role of IoT in enhancing transparency and efficiency through real-time monitoring and data collection. Furthermore, we value your recognition of the challenges and solutions we outlined, such as data heterogeneity, privacy concerns, scalability issues, and regulatory constraints. Your mention of federated learning for secure processing and blockchain for transparent traceability underscores the importance of these emerging technologies in addressing industry challenges. Additionally, we are pleased that our discussion on edge AI and sustainable AI-IoT integration was found to be insightful. We have made some modifications according to the other authors comments. We hope that these changes improve the paper further. Your feedback reinforces the significance of our research and encourages us to further explore these transformative technologies. Thank you once again for your constructive and encouraging remarks.

Reviewer 2 Report
Comments and Suggestions for Authors
Background Theory: The classification is scattered and inaccurate. It is recommended to elaborate the relevant technologies from the perspectives of AI and IoT.
Challenges in food sector using AI: It is suggested to describe the challenging issues and then illustrate the solutions based on state of the art of AI.
Data Sources in Food Computing: It is brief. I suggest elaborating relevant works based on AI and IoT.
The convergence of AI and IoT in food sector: Are there any repetitions with previous contents?
Discussion: I suggest elaborating current problems and clarifying future works.
Overall, it is recommended to supplement necessary comments rather than simply listing related works.
Author Response
|
Comments 1: Background Theory: The classification is scattered and inaccurate. It is recommended to elaborate the relevant technologies from the perspectives of AI and IoT. |
|
Response 1: Thank you for pointing this out. We agree with this comment. In response to your suggestion regarding the classification being scattered and inaccurate, we have elaborated on the relevant technologies from the perspectives of AI and IoT in ‘2.3.1. AI and IoT in Food Sector’. This section now provides a clearer and more structured discussion of these technologies and their role in the food industry. These changes can be found on page 11, paragraph (2.3.1. AI and IoT in Food Sector ), lines (317-322). |
|
Comments 2: Challenges in food sector using AI: It is suggested to describe the challenging issues and then illustrate the solutions based on state of the art of AI. |
|
Response 2: We appreciate this remark and have taken it into account accordingly. Therefore, we have expanded Section 3.2, titled "Challenges in Food Sector Using AI," to provide a more comprehensive discussion on challenges such as data heterogeneity, scalability, privacy concerns, and potential solutions based on state-of-the-art AI advancements. Additionally, we have incorporated relevant references to support our discussion. These changes can be found on pages (6-7), paragraph (3.2. Challenges in food sector using AI), lines (170-207). |
|
Comments 3: Data Sources in Food Computing: It is brief. I suggest elaborating relevant works based on AI and IoT. |
|
Response 3: Thank you for your valuable feedback. We fully agree with this comment. We have, accordingly, expanded Section 4, "Data Sources in Food Computing," to include more relevant works that use datasets in the context of AI and IoT. These references were chosen based on their strong relevance to food computing, their methodological rigor in data collection and analysis, and their contributions to understanding how AI and IoT enhance data utilization in this field. This section now provides a more comprehensive overview of the role of data sources in food computing and their connection to these technologies. These changes can be found on page 19, paragraph (4.1. Types of Data and 4.2. Data Collection Methods), lines (611, 712). |
|
Comments 4: The convergence of AI and IoT in food sector: Are there any repetitions with previous contents? |
|
Response 4: Thank you for bringing this to our attention. We completely recognize and agree with your remark. We have reviewed the manuscript to ensure there is no unnecessary repetition. The Background section provides a high-level introduction to AI and IoT, while the later section elaborates on their transformative impact on the food sector. These sections serve distinct purposes and do not repeat content unnecessarily. |
|
Comments 5: Discussion: I suggest elaborating current problems and clarifying future works. |
|
Response 5: Thank you for your insightful remark. We acknowledge and fully concur with this point. In response, we have expanded the Discussion section after Table 2 to explicitly highlight key unresolved issues such as fragmented data collection, scalability limitations, ethical and privacy concerns, and real-time processing challenges. Additionally, we have outlined specific future research directions, including the development of standardized regulatory frameworks, culturally adaptive AI models, and advancements in edge computing and federated learning for real-time decision-making. These modifications can be found on page 28, paragraph (6. Discussion), lines (848-864). |
|
4. Additional clarifications |
|
We appreciate the reviewer's valuable feedback, which has significantly enhanced the quality of our manuscript. All revisions have been highlighted in the revised version of the manuscript for ease of review. Thank you for your time and consideration. |

Reviewer 3 Report
Comments and Suggestions for Authors
This article provides an overview of the applications of Artificial Intelligence (AI) and the Internet of Things (IoT) in the food industry, particularly their integration in Food Computing. It offers valuable insights and guidance for researchers and practitioners in the field of Food Computing, which is of significant value. However, there are still some issues that need attention, which will help improve the quality of this article.
- We suggest adding the narrative logic of each chapter at the beginning of Chapter 2 and Chapter 3. This would help readers better understand the content of each section. Without this guidance, simply looking at the chapter titles might lead readers to feel that the sections are not well-aligned, which could affect their understanding.
- Many of the images in the article have low layout efficiency and are not aesthetically pleasing. It is recommended to reformat the layout, adjusting the size and position of the images to avoid excessive dispersion, thereby improving the neatness and visual appeal of the page. Additionally, consider using higher-quality images or charts with a consistent style, making them more informative and visually engaging.
- In the section introducing AI and IoT technology applications, you can add some specific technical implementation details, such as algorithm optimization methods, data processing workflows, etc. For example, when discussing AI for food quality inspection, you could elaborate on how techniques like data augmentation and transfer learning are used to improve model accuracy and generalization ability, helping readers better understand and apply these technologies. Additionally, research global internet companies for food or supply chain-related applications, and select some representative real-world cases to demonstrate the successful application of AI and IoT in the food industry in the corresponding chapters. These cases can cover various application scenarios, such as precision irrigation in smart agriculture, quality control in food processing, and real-time monitoring in food logistics. The discussion is not limited to theoretical exploration, such as blockchain traceability technology, or practical business case studies, allowing readers to more intuitively understand the actual effects and value of the technology.
- The content of the article is generally comprehensive, but some parts lack new perspectives or innovative contributions. While the summary of existing work is quite detailed, the discussion on current research trends and future directions is somewhat weak. I recommend that the author include more insights into the future research hotspots in the field, offering some forward-looking perspectives or potential research paths to enhance the academic contribution of the article. We recommend that you introduce a discussion on the latest developments in the AI field. For example, the advancement of generative AI technologies, such as OpenAI's ChatGPT, is bringing numerous opportunities and challenges to the field of Food Computing. These technologies not only provide new methods for data processing and analysis in food science research but also have the potential to drive innovation in areas like personalized nutrition and food safety monitoring. However, despite their great potential, generative AI also faces several challenges in its application, including data privacy, ethical issues, and technical feasibility. Therefore, it is suggested that you add a section in your review to briefly explore the impact of this technology or include a future outlook in the conclusion. This would not only enhance the practical value of your review but also provide readers with more forward-looking insights. Regarding this topic, we suggest that you refer to and cite the article 'Sun L, Liu D, Wang M, et al. Taming Unleashed Large Language Models with Blockchain for Massive Personalized Reliable Healthcare[J]. IEEE Journal of Biomedical and Health Informatics, 2025.,' which summarizes the advantages and disadvantages of generative AI in its current development.
Overall, this paper is a good research review that references ample literature to explore the applications, challenges, and future development directions of AI and IoT in the food industry. Although there are some shortcomings in terms of technical details, recent case studies, and analysis of different companies in practical applications, these do not detract from the overall value of the paper. By supplementing and improving these aspects, the paper will become more complete and practical.
Author Response
|
Comments 1: We suggest adding the narrative logic of each chapter at the beginning of Chapter 2 and Chapter 3. This would help readers better understand the content of each section. Without this guidance, simply looking at the chapter titles might lead readers to feel that the sections are not well-aligned, which could affect their understanding. |
|
Response 1: Thank you for this suggestion. We agree that adding narrative introductions to these chapters would enhance clarity. Therefore, we have added the following narrative logic at the beginning of Chapters 2 and 3 to guide readers on the structure and focus of each section: · Chapter 2: Introduction added to outline the discussion on AI and IoT in food applications. The revised manuscript is updated on page 3, lines (75-89). · Chapter 3: A brief narrative added to explain the detailed exploration of specific use cases and technologies. The revised manuscript is updated on page 10, lines (284-291). |
|
Comments 2: Many of the images in the article have low layout efficiency and are not aesthetically pleasing. It is recommended to reformat the layout, adjusting the size and position of the images to avoid excessive dispersion, thereby improving the neatness and visual appeal of the page. Additionally, consider using higher-quality images or charts with a consistent style, making them more informative and visually engaging.
|
|
Response 2: Thank you for your valuable feedback. We fully agree with this comment. We have improved the visual appeal and layout of the images and we have made the following adjustments: l A new image, Figure 2, was added to improve content representation. These updates can now be found in the revised manuscript on page 4, with a focus on layout improvement. l We have also changed the format of Figure 6 to enhance its clarity and ensure consistency with the overall manuscript design. These updates can be found in the revised manuscript on page 9. l Image 10 has been revised for clarity and visual appeal, ensuring that it is appropriately sized and aligned with the text. These updates can be found in the revised manuscript on page 16, with a focus on layout improvement. |
|
Comments 3: In the section introducing AI and IoT technology applications, you can add some specific technical implementation details, such as algorithm optimization methods, data processing workflows, etc. For example, when discussing AI for food quality inspection, you could elaborate on how techniques like data augmentation and transfer learning are used to improve model accuracy and generalization ability, helping readers better understand and apply these technologies. Additionally, research global internet companies for food or supply chain-related applications, and select some representative real-world cases to demonstrate the successful application of AI and IoT in the food industry in the corresponding chapters. These cases can cover various application scenarios, such as precision irrigation in smart agriculture, quality control in food processing, and real-time monitoring in food logistics. The discussion is not limited to theoretical exploration, such as blockchain traceability technology, or practical business case studies, allowing readers to more intuitively understand the actual effects and value of the technology. |
|
Response 3: Thank you for bringing this to our attention. We completely recognize and agree with your remark. We have expanded the section on AI and IoT technology applications by adding specific technical details. These include discussions on algorithm optimization, data processing workflows, and the use of data augmentation and transfer learning techniques in food quality inspection. These changes can be found on page (6-7), paragraph (2.3.1. AI and IoT in Food Sector 165), lines (170-207). |
|
Comments 4: The content of the article is generally comprehensive, but some parts lack new perspectives or innovative contributions. While the summary of existing work is quite detailed, the discussion on current research trends and future directions is somewhat weak. I recommend that the author include more insights into the future research hotspots in the field, offering some forward-looking perspectives or potential research paths to enhance the academic contribution of the article. We recommend that you introduce a discussion on the latest developments in the AI field. For example, the advancement of generative AI technologies, such as OpenAI's ChatGPT, is bringing numerous opportunities and challenges to the field of Food Computing. These technologies not only provide new methods for data processing and analysis in food science research but also have the potential to drive innovation in areas like personalized nutrition and food safety monitoring. However, despite their great potential, generative AI also faces several challenges in its application, including data privacy, ethical issues, and technical feasibility. Therefore, it is suggested that you add a section in your review to briefly explore the impact of this technology or include a future outlook in the conclusion. This would not only enhance the practical value of your review but also provide readers with more forward-looking insights. Regarding this topic, we suggest that you refer to and cite the article 'Sun L, Liu D, Wang M, et al. Taming Unleashed Large Language Models with Blockchain for Massive Personalized Reliable Healthcare[J]. IEEE Journal of Biomedical and Health Informatics, 2025.,' which summarizes the advantages and disadvantages of generative AI in its current development. |
|
Response 4: Thank you for your insightful remark. We acknowledge and fully concur with this point. In response to your feedback, we have added a new subsection in the manuscript discussing the role of generative AI technologies in advancing Food Computing. This new subsection covers the potential applications of generative AI, such as OpenAI's ChatGPT, in areas like personalized nutrition and food safety monitoring. We also addressed the challenges related to data privacy, ethics, and the technical feasibility of large-scale deployment. Additionally, we discussed the intersection of generative AI and blockchain technology to enhance transparency and security in food traceability. However, after reviewing the suggested paper, we found that it does not have a direct relation to our topic. The paper 'Taming Unleashed Large Language Models with Blockchain for Massive Personalized Reliable Healthcare' primarily focuses on the integration of large language models with blockchain for personalized healthcare applications. While it provides insights into the advantages and challenges of generative AI in the healthcare domain, its scope differs from our work, which is centered on the role of AI and IoT in Food Computing. Our review specifically explores how generative AI can enhance food traceability, personalized nutrition, and food safety monitoring. Therefore, while we appreciate the recommendation, we believe that citing this paper would not be fully relevant to our study. These modifications, including the new subsection, can be found on page 18, paragraph 3.3 (lines 553-594). |
|
4. Additional clarifications |
|
We appreciate the reviewer's valuable feedback, which has significantly enhanced the quality of our manuscript. All revisions have been highlighted in the revised version of the manuscript for ease of review. Thank you for your time and consideration. |

Reviewer 4 Report
Comments and Suggestions for Authors
Aa a review paper, it is suggested that the author focus more on the research status in this field. For those machine learning and data acquisition methods that have been applied to this research topic, the author should review in detail and clarify the strengths and weaknesses of these methods in this research topic. For those machine learning methods that have not yet been applied, the author should indicate potential application directions in this research topic. In this way, it is better to inspire readers.
In addition, redundant text should be removed. For example, in Section 3.2.12, what relationship between the content “The integration of Artificial Intelligence (AI) and the Internet of Things (IoT) has significantly transformed various industries, including the field of food computing. However, these advancements come with intricate and varied challenges, particularly in the food sector. To fully harness AI’s potential to transform food production, safety, and customer experience, these obstacles must be addressed. As the demand for smarter and more efficient food systems grows, there are critical areas where AI and IoT can play pivotal roles.” and the Section title “Standardization and Interoperability of data”? It is suggested that the author check the full manuscript to make the structure of the article more reasonable and the content more prominent.
Author Response
|
Comments 1: Aa a review paper, it is suggested that the author focus more on the research status in this field. For those machine learning and data acquisition methods that have been applied to this research topic, the author should review in detail and clarify the strengths and weaknesses of these methods in this research topic. For those machine learning methods that have not yet been applied, the author should indicate potential application directions in this research topic. In this way, it is better to inspire readers. |
|
Response 1: Thank you for this valuable suggestion. We agree with this comment. In response, we have expanded «2.2.1. AI in Food Sector» to provide a more detailed review of the machine learning and data acquisition methods that have been applied in the food sector. In this revision, we discuss their strengths and weaknesses in the context of food computing. Additionally, we have highlighted machine learning methods that have not yet been widely applied, such as deep reinforcement learning, and outlined potential directions for their application. These revisions aim to provide a clearer view of the current state of research in this field and offer inspiration for future studies. The revised manuscript is updated on pages (4-5), lines (112-135). |
|
Comments 2: In addition, redundant text should be removed. For example, in Section 3.2.12, what relationship between the content “The integration of Artificial Intelligence (AI) and the Internet of Things (IoT) has significantly transformed various industries, including the field of food computing. However, these advancements come with intricate and varied challenges, particularly in the food sector. To fully harness AI’s potential to transform food production, safety, and customer experience, these obstacles must be addressed. As the demand for smarter and more efficient food systems grows, there are critical areas where AI and IoT can play pivotal roles.” and the Section title “Standardization and Interoperability of data”? It is suggested that the author check the full manuscript to make the structure of the article more reasonable and the content more prominent. |
|
Response 2: We appreciate your feedback regarding redundant text and the structural clarity of the manuscript. We agree with this comment. We have reviewed the entire manuscript to remove redundant content, ensuring that the narrative is more concise and focused. Specifically, in Section 3.2.12, we have clarified the content to better align with the section title, "Standardization and Interoperability of Data," and removed the unrelated introductory text. Furthermore, we made additional structural adjustments throughout the article to improve its logical flow, ensuring that the content is both prominent and directly relevant to the research topic. The revised manuscript is updated on page 17, lines (530-545). |
|
4. Additional clarifications |
|
We appreciate the reviewer's valuable feedback, which has significantly enhanced the quality of our manuscript. All revisions have been highlighted in the revised version of the manuscript for ease of review. Thank you for your time and consideration. |

Reviewer 5 Report
Comments and Suggestions for Authors
The authors present a review on an interesting topic for many readers, not only in the food sector. The review consists of 7 chapters including introduction, discussion, and conclusions. The second and the third chapters on background theory and state of the art cotain the most sub chapters in a quite unstructured manner. While Chapter 2 has only a single sublevel with nine subchapters, the third chapter has only two subchapters, but the second consists of twelve subchapters. This imbalance is worsened by the very different topics of the individual subchapters. This structure must definitely be improved for the final version for better clarity.
Some minor issues should be revised before publication:
- Abstract, what is meant with “Food computing”? This expression directly at the beginning needs some explanation. This for the review very important concept is explained much too late in chapter 2.1.
- Figure 1 give the number of publications in the last 5 years in the field of food computing according to Web of Science. The investigated time span should be larger, at least 10-20 years for better statistics. The search items must clearly be named for reproducibility, not all the terms. The text must not repeat the numbers, but illustrate the main trend.
- in Chapter 2.1, the term “food” is defined, however, the review deals with main parts of the food value chain. Are all elements of the value chain treated, such as food storage, industrial food preparation and conservation? A sketch on this entire value chain from seed to consumer would be helpful earlier, not only in Fig. 12.
- As said in the beginning, the structure of chapter 2 as well as 3 has to be revised and should give a better orientation for the reader. Methods must be separated from applications, data handling, or regulatory issues. In chapter 2, consumer needs must be integrated.
- Chapter 3.1 on the current use is quite short, where some of the 3.2 subchapters might fit in. Further, some references are introduced quite generic with “another study [xx]” or “in paper [yy]”. This must be more specific with the given author or institutions. Similar in Chapter 5, please revise.
- Figure 8 mentions elements of “ontology management”, but without reference or specific example. Please add more references to this important topic. The connection to the data types would be helpful.
- Chapters 4.1 and 4.2 give just a list of data types and collections methods, but no introductory text. Please add more explanations for the interested readers.
- Table 2, second column: nearly every entry mentions “is crucial, essential” or “present challenges”, which is typically clear and brings nothing new. Please be more creative and closer to the references in describing the findings.
- References: please check the style for author names, journal names, or publishers.
Author Response
|
Comments 1: Abstract, what is meant with “Food computing”? This expression directly at the beginning needs some explanation. This for the review very important concept is explained much too late in chapter 2.1. |
|
Response 1: Thank you for this suggestion. We agree that the term Food Computing is a central concept in our study and should be introduced earlier. To address this, we have now included a definition of Food Computing in the Abstract, ensuring that readers are familiar with the term from the beginning. The revised manuscript is updated on page 1, lines (1-4). |
|
Comments 2: Figure 1 give the number of publications in the last 5 years in the field of food computing according to Web of Science. The investigated time span should be larger, at least 10-20 years for better statistics. The search items must clearly be named for reproducibility, not all the terms. The text must not repeat the numbers, but illustrate the main trend.
|
|
Response 2: Thank you for this suggestion. We agree that the publication statistics in Figure 1 needed improvement. We have now extended the analysis to cover 10 years instead of 5, providing a more comprehensive view of research trends in Food Computing. Additionally, to enhance transparency and reproducibility, we have explicitly stated the search terms used in Web of Science. Furthermore, the description of Figure 1 has been revised to emphasize key trends rather than simply repeating numerical results. The revised manuscript is updated on pages (2-3), lines (50-63). |
|
Comments 3: in Chapter 2.1, the term “food” is defined, however, the review deals with main parts of the food value chain. Are all elements of the value chain treated, such as food storage, industrial food preparation and conservation? A sketch on this entire value chain from seed to consumer would be helpful earlier, not only in Fig. 12. |
|
Response 3: Thank you for this suggestion. We agree that the discussion on the food value chain was incomplete, as it primarily focused on primary production and consumption. We have now expanded our analysis to include key aspects such as food storage, industrial food preparation, and conservation. To improve clarity, we have also added a new figure (Figure 2) that visually represents the entire food value chain, ensuring a more structured overview. The revised manuscript is updated on page 4, lines (98-107). |
|
Comments 4: As said in the beginning, the structure of chapter 2 as well as 3 has to be revised and should give a better orientation for the reader. Methods must be separated from applications, data handling, or regulatory issues. In chapter 2, consumer needs must be integrated.
|
|
Response 4: Thank you for this suggestion. We agree that Chapter 2 required better organization to distinguish between different aspects of Food Computing. To address this, we have restructured Chapter 2 into clearly defined sections: methods, applications, data handling, regulations, and consumer needs. This ensures that each aspect is discussed in a logical and structured manner. Moreover, we have expanded Chapter 3.1 and integrated relevant content from Chapter 3.2, making the discussion on current applications of Food Computing more comprehensive. |
|
Comments 5: some references are introduced quite generic with “another study [xx]” or “in paper [yy]”. This must be more specific with the given author or institutions. Similar in Chapter 5, please revise. |
|
Response 5: Your feedback is much appreciated. We agree that the citation style needed improvement. We have carefully revised the manuscript to replace generic references like “another study [xx]” with explicit mentions of author names and institutions. Additionally, the references in Chapter 5 have been refined for better clarity and consistency. |
|
Comments 6: Figure 8 mentions elements of “ontology management”, but without reference or specific example. Please add more references to this important topic. The connection to the data types would be helpful. |
|
Response 6: Thank you for this suggestion. We agree that ontology management in Figure 8 needed more specificity. We have now included concrete examples of widely used ontology frameworks, such as Food Ontology (FO) and Agriculture Ontology (AgriOnt). Furthermore, we have expanded the discussion in Section 3.2.1 to clarify how these ontologies contribute to structuring food-related datasets for AI applications. The revised manuscript is updated on page 12, lines (342-357). |
|
Comments 7: Chapters 4.1 and 4.2 give just a list of data types and collections methods, but no introductory text. Please add more explanations for the interested readers. |
|
Response 7: Thank you for your valuable feedback. We agree that Sections 4.1 and 4.2 lacked introductory text, making them harder to follow. To improve readability, we have now added clear introductions to these sections, providing necessary context before presenting the lists of data types and collection methods. |
|
Comments 8: Table 2, second column: nearly every entry mentions “is crucial, essential” or “present challenges”, which is typically clear and brings nothing new. Please be more creative and closer to the references in describing the findings. |
|
Response 8: Thank you for your insightful remark. We acknowledge and fully concur with the point that Table 2 contained repetitive descriptions, which reduced clarity. We have carefully revised the table to improve precision and variation, ensuring that each entry is distinct and informative. The revised manuscript is updated on page 27. |
|
Comments 9: References: please check the style for author names, journal names, or publishers. |
|
Response 9: Thank you for bringing this to our attention. We agree that there were formatting inconsistencies in the reference section. We have now thoroughly checked and reformatted all references to ensure consistency in author names, journal titles, and publisher details. |
|
4. Additional clarifications |
|
We appreciate the reviewer's valuable feedback, which has significantly enhanced the quality of our manuscript. All revisions have been highlighted in the revised version of the manuscript for ease of review. Thank you for your time and consideration. |

Round 2
Reviewer 2 Report
Comments and Suggestions for Authors
In my opinion, the authors have addressed all the previous concerns.
Author Response
Comments 1: In my opinion, the authors have addressed all the previous concerns.
Response 1: We thank the Reviewer for the suggestions and we appreciate the time and effort to review the manuscript, that allowed the authors to improve the final quality of the paper.
We also appreciate that all the previous concerns are considered addressed from the thoroughly review.
Reviewer 3 Report
Comments and Suggestions for Authors
The article primarily discusses the challenges, opportunities, and future directions of artificial intelligence and the Internet of Things in the food sector. During the first round of review, I pointed out that the writing was overly broad and superficial, merely touching on concepts without focusing on the specific challenges of AI and IoT in the food sector. Several key pieces of literature were not cited or discussed. In this round of submissions, the necessary revisions to the paper have still not been made. Especially for a review article, it is crucial to present one's own insights into future directions, which, regrettably, the author has failed to do. Although it is an open-access journal, as a SCI-indexed publication, it is imperative to maintain the quality of the manuscripts. Therefore, I recommend rejection.
Comments on the Quality of English LanguageGood
Reviewer 4 Report
Comments and Suggestions for Authors
The paper can be accepted.
Author Response
Comments 1: The paper can be accepted.
Response 1: We thank the reviewer for previous suggestions and appreciate the time and effort in revising the manuscript, which allowed the authors to improve the final quality of the article. We are pleased that the paper has been proposed for acceptance.
Reviewer 5 Report
Comments and Suggestions for Authors
thank you for the revisions, which make the contribution much clearer and joyful to read
Author Response
Comments 1: thank you for the revisions, which make the contribution much clearer and joyful to read
Response 1: We thank the reviewer for previous suggestions and appreciate the time and effort in revising the manuscript, which allowed the authors to improve the final quality of the article.
We are pleased that the manuscript, in its actual final form, sounds much clearer and joyful to read, as it was in the idea of the authors.
Thanks!